



# Life cycle dynamics of Greenland blocking from a potential vorticity perspective

Seraphine Hauser[1], Franziska Teubler[2], Michael Riemer[2], Peter Knippertz[1], and Christian M. Grams[1]

[1]Institute of Meteorology and Climate Research – Troposphere Research (IMK-TRO), Karlsruhe Institute of Technology (KIT), Karlsruhe, Germany
[2]Institute for Atmospheric Physics, Johannes Gutenberg-University Mainz, Mainz, Germany

**Correspondence:** Seraphine Hauser (seraphine.hauser@kit.edu)

**Abstract.** Blocking over Greenland has substantial impacts on surface weather in particular over Europe and North America, and can increase melting of the Greenland Ice Sheet. Climate models notoriously underestimate the frequency of blocking over Greenland in historical periods, but the reasons for this are not entirely clear, as we are still lacking a full dynamical understanding of Greenland blocking from formation through maintenance to decay. This study investigates the dynamics of

blocking life cycles over Greenland based on ERA5 reanalysis data from 1979–2021. A year-round weather regime definition allows us to identify Greenland blocking as consistent life cycles with an objective onset, maximum, and decay stage. By applying a new quasi-Lagrangian potential vorticity (PV) perspective, following the negative, upper-tropospheric PV anomalies (PVAs$^-$) associated with the block, we examine and quantify the contribution from different physical processes, including dry and moist dynamics, to the evolution of the PVA$^-$ amplitude.

We find that PVAs$^-$ linked to blocking do not form locally over Greenland but propagate into the region along two distinct pathways (termed "upstream" and" retrogression") during the days before the onset. Remarkably, the development of PVAs$^-$ differs more between the pathways than between seasons. Moist processes play a key role in the amplification of PVAs$^-$ before the onset and are linked to midlatitude warm conveyor belts. Interestingly, we find moist processes supporting the westward propagation of retrograding PVAs$^-$ from Europe, too, previously thought to be a process dominated by dry-barotropic Rossby

wave propagation. After onset, moist processes remain the main contribution to PVA$^-$ amplification and maintenance. However, moist processes weaken markedly after the maximum stage and dry processes, i.e. barotropic, non-linear wave dynamics, dominate the decay of the PVAs$^-$ accompanied by a general decrease in blocking area. Our results corroborate the importance of moist processes in the formation and maintenance of Greenland blocking, and suggest that a correct representation of moist processes might help reducing forecast errors linked to blocking in numerical weather prediction models and blocking biases

in climate models.

## 1  Introduction

Atmospheric blocking describes a flow configuration in mid and high latitudes with a dominant, stationary, and long-lived anticyclone (often referred to as 'the block') (e.g., Rex, 1950). It interrupts the usual zonal flow in the upper troposphere and induces a strong meridional flow along the block's edges. A persistent blocking pattern can therefore hinder the eastward



progression of synoptic-scale extratropical weather systems, and can trigger extreme weather events (Kautz et al., 2022). Some of these events in the past include the European cold spell in winter 2009/2010 (Cattiaux et al., 2010), the heatwave in Eastern Europe and Russia in summer 2010 (Grumm, 2011), and the North American heat wave in July 2021 (Oertel et al., 2023).

In the Northern Hemisphere, blocking develops primarily close to the jet stream's exit zones (e.g., Woollings et al., 2018; Lupo, 2021). Despite a large range of identification methods, the majority of methods agree on the two prominent hot spots over the eastern North Atlantic and the eastern North Pacific (Pinheiro et al., 2019). Although blocking over Greenland takes place less frequently compared to blocking over the eastern North Atlantic, it stands out due to its longevity compared to blocking in other regions in the Northern Hemisphere (Drouard et al., 2021). Greenland blocking is more likely to temporarily shift the westerly flow equatorward instead of completely decelerate and block it (Woollings et al., 2008) and is strongly anti-correlated with the North Atlantic Oscillation (NAO), which represents the most prominent pattern of climate variability in the extratropical North Atlantic region (e.g., Wallace and Gutzler, 1981). The presence of a blocking high pressure system over Greenland promotes reduced cloud cover and increased temperatures near the surface, which causes melting of the Greenland Ice Sheet and consequently contributes to the global rise in sea surface level (Rowley et al., 2020; Hermann et al., 2020; Hanna et al., 2021). Details of these impacts depend on the exact block position relative to Greenland's topography, the direction of propagation, the blocked large-scale circulation pattern, and the time of the year (Barrett et al., 2020; Tedesco and Fettweis, 2020; Preece et al., 2022; Pettersen et al., 2022). Furthermore, blocking over Greenland also leads to impacts beyond the blocking region: Arctic sea ice decline and Eurasian cold spells (Chen and Luo, 2017), increased precipitation in the Northeast United States (Simonson et al., 2022), periods of widespread low production of wind and solar power and high electricity demand (Otero et al., 2022; Mockert et al., 2023), and increased winds over southwest Europe with the potential of extreme weather events in this area (Grams et al., 2017; Hauser et al., 2023a).

As blocking over Greenland can cause far-reaching impacts, it is of high importance to accurately predict its occurrence in advance. Despite many model improvements, current numerical weather prediction models still underestimate the frequency of blocking, in particular over the North Atlantic (Quinting and Vitart, 2019; Davini and D'Andrea, 2016). Although Greenland blocking has a higher prediction skill compared to blocking over the eastern North Atlantic and Europe (Büeler et al., 2021; Hochman et al., 2021; Osman et al., 2023), the reasons for the underestimation of blocking in climate models for the historical period are not entirely clear yet. A better understanding of the biases in physical mechanisms is necessary to improve the representation of blocking, and, in particular, to correctly predict possible changes in blocking over Greenland in future scenarios (Michel et al., 2021).

The dynamics of atmospheric blocking have been investigated from different angles, and the review articles of Woollings et al. (2018) and Lupo (2021) provide good state-of-the-art synopses on different blocking theories. Only a few studies explicitly analyze the processes linked to blocking over Greenland, and often discuss the insights in conjunction with blocking over the North Atlantic (e.g., Martineau et al., 2022), or in terms of the negative phase of the North Atlantic Oscillation (NAO-) (e.g. Rivière and Drouard, 2015), which is closely related to blocking over Greenland (e.g., Woollings et al., 2010). The breaking of upper-level Rossby waves (further referred to as RWB) has been found as a formation mechanism for blocking over Greenland with predominantly cyclonic RWB towards Greenland (Benedict et al., 2004; Woollings et al., 2008; Michel and





Rivière, 2011). Wave-train signals were found in the large-scale evolution before Greenland blocking events (Cheung et al., 2023), and ensemble sensitivities were revealed with high sensitivities to the upper-tropospheric large-scale pattern on low-frequency scales (Parker et al., 2018). However, Michel et al. (2021) found that cyclonic RWB in climate models is not the only mechanism to blocking over Greenland suggesting that the formation of Greenland blocking cannot be explained by dry upper-level wave dynamics alone. By investigating the impact of baroclinic energy conservation to blocking, Martineau et al. (2022)

found that blocks over Greenland belong to the most baroclinic blocks. Low-level baroclinicity provides favorable conditions for the development of extratropical cyclones, which were shown to play a role for blocking development and maintenance (Nakamura and Wallace, 1993; Hwang et al., 2020). For the Greenland region, in particular, McLeod and Mote (2015) revealed that (multiple) precursor cyclones were linked to the intensification of blocking in summer. Although not applied to blocking in Greenland explicitly, further theoretical concepts have provided novel insights into the dynamics of blocking (e.g., Shutts,

1983; Yamazaki and Itoh, 2013; Nakamura and Huang, 2018; Luo et al., 2019). All the studies mentioned above refer more to dry dynamical processes. Schwierz (2001) was one of the first studies investigating in detail the mutual interaction of the Greenland topography with the atmospheric flow and, most notably, they firstly described a substantial contribution of cloud-diabatic processes to the evolution of blocking over the region. Subsequently, multiple studies unveiled latent heat release in ascending air streams as a first order process in establishing the block for single case studies but later on from a climatological

point of view (e.g., Croci-Maspoli and Davies, 2009; Pfahl et al., 2015; Steinfeld and Pfahl, 2019). With a particular focus on Greenland, strong moisture fluxes were found in advance of extreme blocking by Barrett et al. (2020), indicating an important role in developing or sustaining blocks in this region. To link the importance of moist processes to the predictability of blocking, Wandel (2022) recently showed that models systematically underestimate moist processes in the vicinity of Greenland blocks during periods of bad forecast skill, which raises the question regarding the importance of moist processes relative to

the dry dynamics.

Teubler et al. (2023) presented a first investigation on the relative importance of dry and moist dynamics in the formation of blocking from a local potential vorticity (PV) perspective. Using mid-latitude PV thinking (Hoskins et al., 1985), the role of quasi-barotropic dynamics, baroclinic interaction, and the impact of moist processes can be quantified separately, based on the piecewise PV tendency framework of Teubler and Riemer (2016). By extending previous Eulerian tendency approaches used

to describe the evolution of flow patterns (e.g., Feldstein, 2002; Michel and Rivière, 2011), PV tendencies were projected onto and weighted by the mean blocking pattern over Greenland. In this Eulerian perspective of Teubler et al. (2023) on the local emergence of blocking over Greenland, dry dynamics – linear quasi-barotropic dynamics and eddy flux convergence – dominate the formation of blocking over Greenland. Baroclinic interaction and moist processes linked to divergent amplification are diagnosed to be of minor importance. However, other studies point to a key role of latent heating for blocking formation and

maintenance from a Lagrangian perspective (e.g., Pfahl et al., 2015; Steinfeld and Pfahl, 2019). While the PV perspective is in general able to capture moist-baroclinic growth, Teubler et al. (2023) discuss the limitations of the *Eulerian* perspective in capturing the importance of these processes: Moist-baroclinic growth tends to occur in regions where the amplitude of the regime pattern is small, and thus hardly contribute to the tendencies projected onto the pattern. Hauser et al. (2023b) explicitly demonstrate this deficiency of the Eulerian perspective in a case study and reconcile the seemingly contrasting





results on the importance of moist-baroclinic processes by adopting the Eulerian perspective of Teubler et al. (2023) into a new quasi-Lagrangian perspective. Previous studies have partly used quasi-Lagrangian approaches to investigate the role of eddy feedback on blocks and, in particular, the role of transient anticyclonic eddies (e.g., Shutts, 1983; Yamazaki and Itoh, 2013; Suitters et al., 2023). By tracing back negative, upper-tropospheric PV anomalies linked to a block over Europe and investigating the relative contributions of dry and moist dynamics using the same piecewise PV tendencies of Teubler

et al. (2023), the quasi-Lagrangian approach of Hauser et al. (2023b) revealed a non-local development of the negative PV anomaly over the western North Atlantic. This negative PV anomaly propagated eastward and was pulse-like amplified by moist processes over the North Atlantic – a development the Eulerian perspective missed as the PV anomaly was located and associated processes beyond the blocking region which the Eulerian perspective focuses on. Based on these results for a single case study, differences are expected in the importance of dry and moist processes for the formation of blocking over Greenland

when applying the quasi-Lagrangian perspective climatologically and comparing it to the insights of Teubler et al. (2023).

In this study, we use an advanced version of the quasi-Lagrangian perspective of Hauser et al. (2023b) to gain climatological insights into the processes and dynamics of blocking over Greenland. Periods of Greenland blocking are identified from the perspective of weather regimes, which are large-scale persistent, quasi-stationary, and recurrent flow patterns (Vautard, 1990; Michelangeli et al., 1995). Using Greenland blocking as one of the seven year-round weather regimes in the North Atlantic-

European region (Grams et al., 2017), and with an objective regime life cycle definition, it allows for a systematic analysis of the full Greenland Blocking life cycles from the onset over the maximum to the decay during 1979-2021. The purpose of this study is to shed light on the dynamics of blocking over Greenland with the overall goal of a broader understanding on how blocking in this region forms, is maintained, and finally decays. With the insights obtained in this study, the representation of blocking in numerical weather prediction and climate models could be improved by focusing on the processes that shape

blocking over Greenland.

The paper is organized as follows: Section 2.1 introduces the data sets and the further advanced quasi-Lagrangian PV framework originally developed in Hauser et al. (2023b). The analysis on the formation of Greenland blocking is presented in Section 3. Insights into the maximum and decay stages of Greenland Blocking follow in Section 4 to complete a full picture of Greenland Blocking life cycle dynamics. Section 5 provides a summarizing discussion and concluding remarks.

## 2   Data and methods

### 2.1   Data

This study is based on ERA5 reanalysis data (Hersbach et al., 2020) from the European Centre of Medium-Range Weather Forecasts (ECMWF) for the period January 1979 – December 2021. The data set is remapped from the original T639 spectral resolution to a regular latitude-longitude grid. For the identification of negative upper-tropospheric PV anomalies, model level

data is used with a spatial resolution in the horizontal of $0.5\,^{\circ}$ and a temporal resolution of three hours. Spatially coarser data in the horizontal ($1\,^{\circ}$) is needed for the piecewise PV inversion (see Teubler and Riemer, 2021).





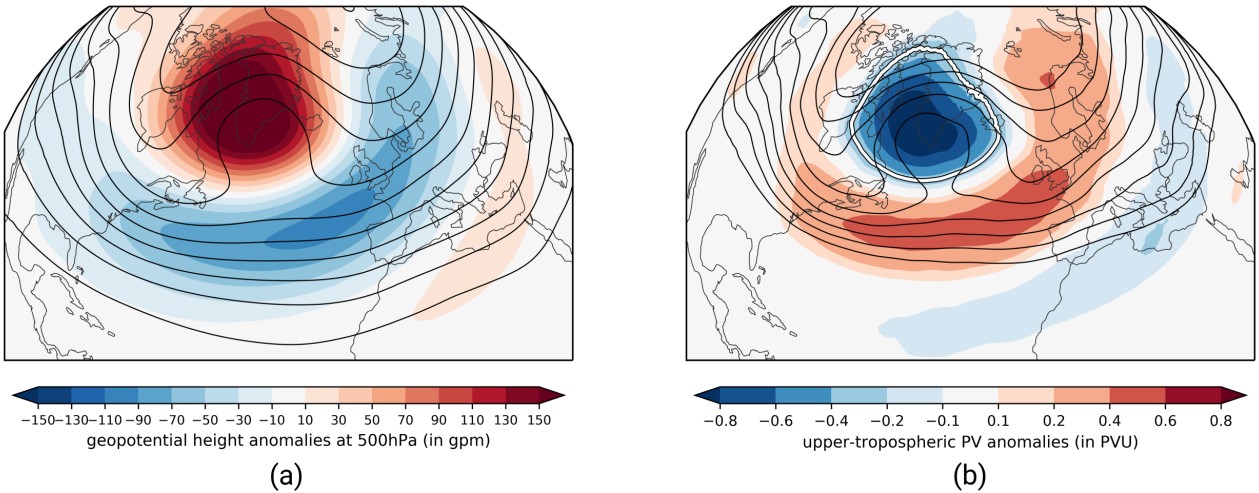

**Figure 1.** (a) Geopotential height at 500 hPa (black lines, from 5250 to 5850 gpm in steps of 60 gpm) and corresponding anomalies (shading, in gpm) and (b) vertically-averaged PVAs between 500–150 hPa (shading, in PVU) and vertically-averaged PV (contours, from 1.5 to 3.5 PVU in steps of 0.25 PVU) for all time steps attributed to the GL regime type. The white solid line in panel (b) illustrates the regime mask for the GL regime type, defined by the -0.3 PVU contour.

## 2.2 Greenland Blocking from a weather regime perspective

As mentioned in the introduction, we here define periods of Greenland blocking from the perspective of weather regimes. Grams et al. (2017) developed a year-round weather regime classification in the North Atlantic-European region (80 °W – 40 °E, 30 °– 90 °N), which is translated from the former ERA-Interim reanalysis of ECMWF (Dee et al., 2011) to ERA5. Weather regimes are identified in the period 1979–2019, and therefore, six-hourly anomalies of geopotential height at 500 hPa (based on a 90-day running mean climatology, 1979–2019) are filtered by a 10-day low-pass filter (Lanczos filter Duchon, 1979). Anomalies are normalized to aim for a year-round definition, and *k*-means clustering is performed for the expanded phase space of the leading seven empirical orthogonal functions that explain 74.4 % of the variability. This definition yields in total seven weather regimes, with three cyclonic (Zonal regime, Scandinavian Trough, Atlantic Trough) and four anticyclonic regime types (Atlantic Ridge, European Blocking, Scandinavian Blocking, Greenland Blocking). In accordance with the regime abbreviations introduced in Grams et al. (2017), we use GL as abbreviation for Greenland Blocking from now on.

Following Michel and Rivière (2011) and Grams et al. (2017), a weather regime index $I_{WR}$ is computed to make a quantitative statement about the similarity of an instantaneous geopotential height field to one of the seven weather regimes. It is defined as

$$I_{WR}(t) = \frac{P_{WR}(t) - \overline{P_{WR}}}{\sqrt{\frac{1}{NT}\sum_{t=1}^{NT}[P_{WR}(t) - \overline{P_{WR}}]^2}} \qquad \text{with} \qquad P_{WR}(t) = \frac{\sum_{(\lambda,\varphi)\in EOF} \Phi^L(t,\lambda,\varphi)\Phi_{WR}^L(\lambda,\varphi)\cos\varphi}{\sum_{(\lambda,\varphi)\in EOF}\cos\varphi}, \qquad (1)$$




where NT is the total number of time steps within a climatological sample (all times in 1979–2019) and $(\lambda, \varphi)$ is the respective longitude/latitude within the EOF domain. $P_{WR}(t)$ is a scalar measure that describes the projection of the filtered anomaly $\Phi^L(t, \lambda, \varphi)$ to the EOF cluster mean $\Phi_{WR}^L(\lambda, \varphi)$ within the EOF domain. $\overline{P_{WR}}$ is the climatological mean of the projection

$P_{WR}$ such that $I_{WR}$ is computed as the deviation of $P_{WR}(t)$ from $\overline{P_{WR}}$ normalized by the standard deviation. Even though the weather regimes are defined based on the 1979–2019 data period, $I_{WR}$ can also be computed beyond this data period for each of the seven regime and each three-hourly time step in the ERA5 period considered (1979–2021).

Based on $I_{WR}$, objective weather regime life cycles and associated life cycle stages are obtained following Grams et al. (2017). Local maxima of $I_{WR}$ with $I_{WR} \geq 1.0$ are determined as preliminary maximum stage of possible weather regime

life cycles. Second, preliminary onset and decay dates are defined as first and last time steps around all maximum stages, where $I_{WR} \geq 1.0$. Finally, regime life cycles are defined as periods bounded by an onset and a decay time, if the difference $\Delta t = t_{decay} - t_{onset}$ amounts to at least five days in order to ensure sufficient persistence of the regime. In case two local maximum stages of the same regime share the same onset or decay time, two regime life cycles are combined if the additional following conditions apply: (i) the mean $I_{WR}$ between the two maximum stages is $\geq 1.0$, and (ii) the time difference between

the two maxima $\Delta t_{max} = t_{max1} - t_{max2} \leq 100$ days. The combined regime life cycle is then characterized by the earliest *onset* and latest *decay* time. The *maximum stage* corresponds to the time, when $I_{WR}$ is highest during the new life cycle period. This definition of weather regime life cycles allows more than one regime to be active at the same time, where 'active' here means that the $I_{WR}$ for more than one regime exceeds 1.0 for at least five days. For strong and meaningful regime life cycles, it applies that the regime must have the highest $I_{WR}$ out of all seven regimes for at least one time step in the active regime life

cycle, such that an in-depth analysis of life cycle stages (onset, maximum, decay) is possible.

177 GL life cycles are identified during 1979–2021, 31 in December–February (DJF), 58 in March–May (MAM), 52 in June–August (JJA), and 36 in September–November (SON). Despite the differences in numbers, the share of days within a season linked to an active GL life cycle is nearly constant (not shown) due to strongly variable length ranging from five days to more than a month (Figure A1a). Figure 1a shows the year-round 500 hPa geopotential height (Z500) pattern during GL.

Positive Z500 anomalies and northward bulging isohypses (black lines) indicate the location of the ridge over Greenland, which is flanked to the south by negative Z500 anomalies as the result of the southward shift of the storm track (cf. Woollings et al., 2008). This pattern resembles the negative phase of the NAO (Feldstein, 2003), and, in accordance with Woollings et al. (2008), there is a high negative correlation between the $I_{WR}$ of GL and the NAO index (Figure 2).

## 2.3  A quasi-Lagrangian PV framework

The quasi-Lagrangian PV framework has been introduced in Hauser et al. (2023b) for a single regime life cycle case study and is here further developed for the systematic investigation of year-round GL regime life cycles.

### 2.3.1  Tracking of upper-tropospheric PV anomalies linked to Greenland Blocking regime life cycles

Analogous to Hauser et al. (2023b), we look at negative upper-tropospheric PV anomalies as vertically-averaged PV between 500 and 150 hPa based on model level data of ERA5, which is consistent with the atmospheric blocking identification algo-





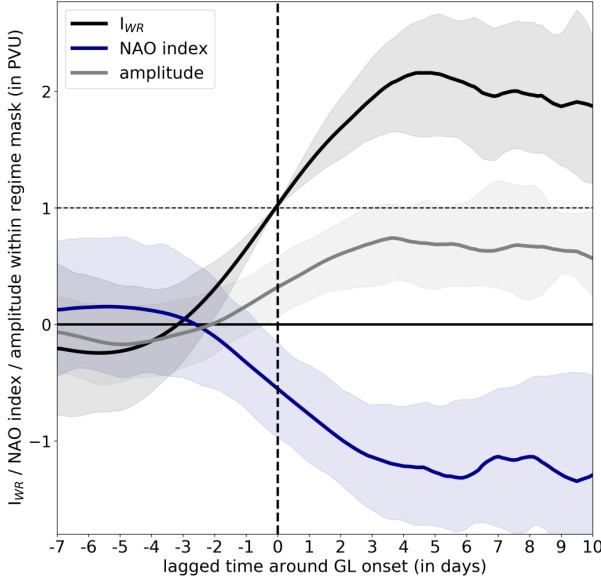

**Figure 2.** Temporal evolution of the weather regime index $I_{WR}$ for GL (black solid), the NAO index (blue solid), and the amplitude of the mean upper-tropospheric PV anomaly (in PVU) within the regime mask of GL (cf. Figure 1b, white contour). Shading indicates the 20–80 percentile range. The horizontal dashed line (in black) marks the $I_{WR}$ threshold used for the definition of regime life cycles.

rithm of Schwierz et al. (2004). Anomalies are calculated as deviations from a 30-day running mean climatology based on the period 1979–2019. Figure 1b displays the mean upper-tropospheric PV anomalies for active GL life cycles. As for the European Blocking regime (Hauser et al., 2023b, their Figure 2), a good agreement between the Z500-based pattern and upper-tropospheric PV field is evident, which justifies a consideration of weather regime dynamics from a PV perspective. As the GL pattern (Figure 1b) indicates the presence of negative upper-tropospheric PV anomalies, we define the area with PV anomalies smaller than -0.3 PVU (white contour) as the regime mask of GL. Thus, in this study, we consider solely PV anomaly objects that spatially overlap this region during the life cycle with a minimum coverage of the regime mask of at least 10 %.

We identify and track anticyclonic anomaly objects of upper-tropospheric PV in agreement to Hauser et al. (2023b). For the European blocking in March 2016, Hauser et al. (2023b) used a fixed threshold for the identification of negative PV anomaly objects, referred to as PVAs⁻. However, for a year-round consideration, a variable threshold is required that depends on the time of the year to account for the fact that PVAs⁻ are stronger in winter than in summer (Steinfeld and Pfahl, 2019). For the European Blocking case in March 2016, Hauser et al. (2023b) used a threshold of -0.8 PVU, which captures approximately the 35 % strongest negative PV anomalies in terms of area in the Northern Hemisphere during 1979–2019. Here, for each calendar day, we determine the threshold required to capture the strongest 35 % of PVAs⁻ for this day. A Fast Fourier Transformation is applied for smoothing and yields the final threshold for every calendar day in the year (Figure A1c). The running threshold shows a maximum in strength around March and a minimum in strength around July. PVAs⁻ are traced in space and time based on the method of Schwierz et al. (2004) with major adjustments (see Hauser et al., 2023b, their Figure A1). In general,





PVAs$^-$ are tracked based on spatial overlap without a criterion on minimum overlap. A further development of the algorithm allows the detection of splitting and merging events along a tracked PVA$^-$. This enables an analysis which examines the role of transient anticyclonic anomalies in feeding a block and is strongly inspired by the work of Shutts (1983), Yamazaki and Itoh (2013) and Suitters et al. (2023).

### 2.3.2 Quantification of processes to the PV anomaly amplitude evolution

Following Hauser et al. (2023b), we apply the piecewise PV tendency framework originally developed for Rossby wave packets (RWPs) by Teubler and Riemer (2016) to investigate the amplitude evolution of PVAs$^-$ identified and traced from a quasi-Lagrangian perspective (cf. Section 2.3.1). We use the Ertel PV definition (Ertel, 1942) on isentropic surfaces as $q = \sigma^{-1}(\zeta_\theta + f)$ considering the hydrostatic approximation, with $\sigma = -g^{-1}(\partial p/\partial \theta)$ as the isentropic layer density with gravity $g$, pressure $p$, and potential temperature $\theta$, $\zeta_\theta$ as relative vorticity on isentropic surfaces, and $f$ the Coriolis parameter. The PV tendency equation describes the change in PV at a fixed point by (i) the advection of PV, and (ii) non-conservative PV modification ($\mathcal{N}$), and reads as

$$\frac{\partial q}{\partial t} = -\boldsymbol{v} \cdot \boldsymbol{\nabla}_\theta q + \mathcal{N}, \tag{2}$$

with the horizontal wind field $\mathbf{v} = (u, v, 0)$ and the gradient operator along an isentropic surface $\boldsymbol{\nabla}_\theta$. Following Teubler and Riemer (2016), the advective PV tendency ($-\boldsymbol{v} \cdot \boldsymbol{\nabla}_\theta q$) is further split into different terms, with each term referring to specific processes in mid-latitude dynamics. Note that – equivalent to above – PV anomalies $q'$ on isentropic surfaces are defined as deviations from a 30-day running mean climatology $q_0$ (1980–2019). The background wind field $\mathbf{v}_0$ is constructed similarly as $q_0$. A Helmholtz partitioning of $\mathbf{v}$ is performed to extract the divergent wind field $\mathbf{v}'_{div}$, and the piecewise PV inversion under nonlinear balance (Davis and Emanuel, 1991; Davis, 1992) yields the non-divergent wind components associated with upper-tropospheric and lower-tropospheric PV anomalies, resulting in the wind fields $\mathbf{v}'_{low}$ and $\mathbf{v}'_{up}$, respectively. All wind fields are interpolated to isentropic levels ranging from 315 to 355 K in steps of 5 K. Finally, the full wind field $\mathbf{v}$ reads as:

$$\mathbf{v} = \mathbf{v}_0 + \mathbf{v}'_{div} + \mathbf{v}'_{up} + \mathbf{v}'_{low} + \mathbf{v}'_{res}, \tag{3}$$

where $\mathbf{v}'_{res}$ is the residual wind field that arises due to (i) characteristics inherent in piecewise PV inversion (e.g., nonlinearities and imperfect knowledge of boundary conditions), (ii) numerical inaccuracies, and (iii) the interpolation of wind fields from pressure to isentropic levels. Analogously to Hauser et al. (2023b), the amplitude metric as spatial integral of $q'$ over the PV anomaly area $A(t)$ is given by

$$\frac{d}{dt} \int_{A(t)} q' dA = \int_{A(t)} \frac{\partial q}{\partial t} dA - \int_{A(t)} \frac{\partial q_0}{\partial t} dA + \oint_{S(t)} q'(\mathbf{v}_s \cdot \mathbf{n}) dS, \tag{4}$$

with $S(t)$ the boundary of $A(t)$, $\mathbf{v}_s$ the motion of the boundary $S(t)$, and the normal vector $\mathbf{n}$. Using the decomposition of the wind field and further transformations thoroughly documented in Appendix A, the final equation for the amplitude evolution





of PV anomalies reads as

$$\frac{d}{dt}\int_{A(t)} q'\,dA = -\int_{A(t)} \mathbf{v}'_{up}\cdot\boldsymbol{\nabla}q_0\,dA - \underbrace{\int_{A(t)} <-\mathbf{v}'_{up}\cdot\boldsymbol{\nabla}q_0>\,dA}_{UP} - \underbrace{\int_{A(t)} \mathbf{v}'_{low}\cdot\boldsymbol{\nabla}q_0\,dA - \int_{A(t)} <-\mathbf{v}'_{low}\cdot\boldsymbol{\nabla}q_0>\,dA}_{LOW}$$

$$\underbrace{-\int_{A(t)} \mathbf{v}'_{div}\cdot\boldsymbol{\nabla}q_0\,dA - \int_{A(t)} <-\mathbf{v}'_{div}\cdot\boldsymbol{\nabla}q_0>\,dA}_{DIV_{adv}} + \underbrace{\int_{A(t)} q'(\boldsymbol{\nabla}\cdot\mathbf{v}'_{div})\,dA - \int_{A(t)} <q'(\boldsymbol{\nabla}\cdot\mathbf{v}'_{div})>\,dA}_{DIV_{div}}$$

$$\underbrace{-\int_{A(t)} \mathbf{v}'_{res}\cdot\boldsymbol{\nabla}q_0\,dA - \int_{A(t)} <-\mathbf{v}'_{res}\cdot\boldsymbol{\nabla}q_0>\,dA}_{RES} + \underbrace{\int_{A(t)} \mathcal{N}\,dA - \int_{A(t)} \mathcal{N}_0\,dA}_{NONCONS}$$

$$\underbrace{+\oint_{S(t)} q'(\mathbf{v}_s - \mathbf{v})\,d\mathbf{S}}_{\mathcal{B}nd}, \tag{5}$$

with <> as a mean operator that consists of averages between 1980–2019 for each calendar day and a subsequent running mean
(+/- 15 days). The individual PV tendency terms (abbreviations in Equation 5 are to be interpreted as follows. The term UP
is closely related to barotropic dynamics, as it represents the advection of upper-tropospheric PV by the wind field associated
with upper-tropospheric PV anomalies. The modification of upper-tropospheric PV anomalies by the wind fields linked to
lower-tropospheric PV anomalies is described by LOW and reflects baroclinic interaction of lower levels with upper levels.
The change in $q'$ by the divergent wind field $\mathbf{v}'_{div}$ is governed by (i) the advection of background PV with the divergent wind
field ($DIV_{adv}$), and (ii) the divergence of the divergent wind within the PV anomaly ($DIV_{div}$), which often applies to a change
in the area of the PV anomaly. Upper-tropospheric divergent outflow is often linked to latent heat release below, such that both
terms – $DIV_{adv}$ and in particular $DIV_{div}$ – have often been referred to as indirect moist-dynamical contributions (e.g., Riemer
et al., 2008; Grams et al., 2011; Steinfeld and Pfahl, 2019). In midlatitudes, latent heat release occurs predominantly within
ascending air streams, which are known as warm conveyor belts (WCBs; Wernli, 1997) and occur in the vicinity of extratropical
cyclones (Madonna et al., 2014; Pfahl et al., 2015). Thus the terms $DIV_{adv}$ and $DIV_{div}$ also include the role of divergent WCB
outflow in the upper troposphere. The term RES describes the advection of background PV by the residual wind field and is
hard to describe in a physical sense, as it includes the part of the framework that cannot be explained by the wind fields obtained
from Helmholtz partitioning and piecewise PV inversion. The modification by non-conservative processes is further referred
to as NONCONS. Previous studies have shown that this term is of minor importance for the amplitude evolution of ridges and
240 troughs since small-scale non-conservative PV tendencies often are too localized or cancel out integrated over the anomaly
surfaces (Teubler and Riemer, 2016; Hauser et al., 2023b). Analogously to Teubler et al. (2023), we do not explicitly consider
NONCONS in this study. The diagnosed amplitude change (DIAG) in Equation 5 changes to DIAG = UP + LOW + $DIV_{div}$ +
$DIV_{ADV}$ + RES + $\mathcal{B}$nd.

Equation 5 is evaluated on isentropic surfaces, with the isentropic level depending on the season. Following Röthlisberger
et al. (2018), we use 320 K for December, January, February, March; 325 K in April, November; 330 K in May, October; 335 K



in June, September; and 340 K in July, August. Analog to Teubler et al. (2023), we average values within +/- 5 K around the selected isentropic surface. DIAG is compared to the observed amplitude change (OBS), the latter of which is calculated from the forward difference of the area-integrated PV anomaly amplitude between two time steps. Further information on the estimation of $\mathcal{B}nd$ and reasons for deviations between DIAG and OBS are provided in Appendix B.

## 2.4 Eulerian identification of WCBs

As the identification of WCBs with Lagrangian air parcel trajectories is associated with rather expensive computations for the large ERA5 period, we use the novel EuLerian Identification of ascending AirStreams (ELIAS 2.0) data set, which has been developed by Quinting and Grams (2022). ELIAS 2.0 uses convolutional neural networks (CNNs) fed with instantaneous gridded fields to predict footprints of different WCB stages: WCB inflow in the lower troposphere, WCB ascent in the mid-troposphere, and WCB outflow in the upper troposphere. For each WCB stage, the four most important predictors identified from a step-wise forward selection approach (Quinting and Grams, 2021) are used as input data. For a year-round WCB identification, the 30-day running mean Lagrangian-based WCB climatology serves as fifth predictor. After training the CNNs based on 12-hourly ERA-Interim data (Dee et al., 2011) from 1980—1999, the CNNs are applied to 3-hourly ERA5 data and provide a conditional probability for WCBs (ranging from 0 to 1). A grid-point dependent decision threshold decides if a certain probability is associated with any of the three WCB stages. This threshold is determined for each day of the year and for each WCB stage, such that the bias between the Lagrangian-based and CNN-based WCB climatology is minimal.

## 3 Formation of Greenland Blocking

This section provides insights into the onset dynamics of GL. Therefore, we here apply the quasi-Lagrangian PV framework (Section 2.3) to all GL regime life cycles in the considered ERA5 period. Specifically we investigate the origin of PVAs$^-$ linked to blocking and disentangle the contribution of dry and moist processes in the life cycle of blocked regimes.

### 3.1 Large-scale PV evolution

First we explore the formation of the GL pattern from an Eulerian PV perspective. Figure 3 (upper row) shows the development of GL regime life cycles centered on all GL onset dates between 1979–2021 based on the field of upper-tropospheric PV anomalies, mean sea level pressure (MSLP), and Z500.

The onset of GL is characterized by a rapid emergence of a blocked situation out of a very zonally-oriented circulation pattern over the western North Atlantic within about six days (Figure 3, upper row). A weak and broad trough prevails over the western North Atlantic six days before the onset (Figure 3a). No clear upstream wave-train signal is present, which matches the observation of Feldstein (2003) that the negative NAO phase often develops in-situ. Low MSLP over the western North Atlantic indicates the presence of extratropical cyclones downstream of the trough close to Greenland in the days before the onset (blue contours in Figure 3a,b), which suggests an involvement of synoptic moist-baroclinic activity over the central North Atlantic to the development of GL. The circulation pattern over the western North Atlantic hardly changes in the days before





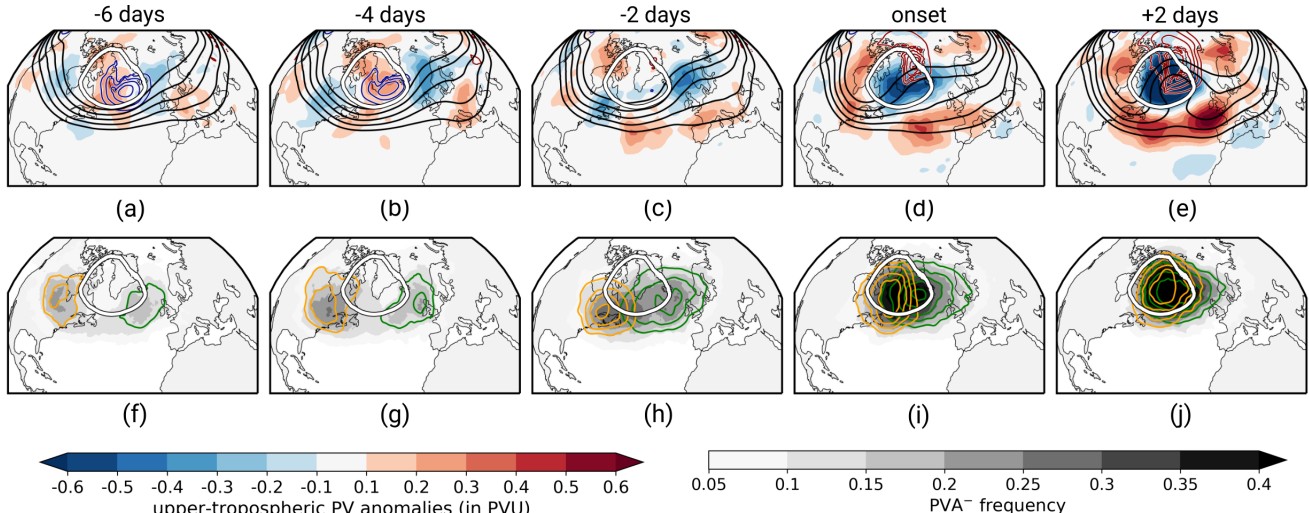

**Figure 3.** Large-scale year-round PV evolution of GL. Top: Upper-tropospheric PV anomalies measured as vertically-averaged mean between 150 and 500 hPa (cf. Section 2.3.1, in PVU, shading) and Z500 (black solid contours, levels: 53500, 54000, 54500, 55000, 55500 gpm) for selected time steps relative to GL onset. Blue and red contour show mean sea level pressure (blue: 1000, 1002, 1004, 1006, 1008 hPa; red: 1020, 1022, 1024, 1026, 1028 hPa). Bottom: Frequency of onset PVAs⁻ (grey shading) and frequencies of onset PVAs⁻ following the upstream pathway (orange contours) and the retrogression pathway (green contours). The white contour represents the regime mask for GL (cf. Figure 1b). Note that for each time lag, the time steps +/- 6 hours were taken into account.

the onset (Figure 3a–c). In contrast, the circulation pattern downstream of Greenland indicates ridge building over Europe in the days before the onset. Previous studies highlight Scandinavian blocking or Atlantic ridge as precursor regime patterns to blocking over Greenland or to negative phases of the NAO (e.g., Vautard, 1990; Cassou, 2008; Michel and Rivière, 2011; Luo et al., 2012; Büeler et al., 2021), which is consistent with high $I_{WR}$ values of AR, EuBL and ScBL in the days before GL onset (see Figure A3). The most rapid development of the large-scale pattern over Greenland takes place in the four days around regime onset (Figure 3c–e), characterized by a fast amplification of the ridge that establishes over Greenland and is associated with strong negative PV anomalies in the upper troposphere. In accordance with Woollings et al. (2010), a split flow is visible from the onset on (Figure 3d,e), pointing to the typical southward shift of the midlatitude storm track during GL (cf. Figure 1b). Strong negative PV anomalies prevail over Greenland and high MSLP values show that the anticyclonic circulation in the upper troposphere over Greenland has manifested as a high pressure anticyclone near the surface (high MSLP, red contours).

## 3.2 Pathways of PVAs⁻ to Greenland

Two separate regions of negative PV anomalies, namely over the northeastern United States and over northern Europe, stand out in the days before GL onset (Figure 3b,c), suggesting a propagation of anomalies rather than a rapid in-situ development of anomalies over Greenland. Croci-Maspoli et al. (2007) found that the circulation anomaly linked to the ridge over Europe propagated westward towards to build up a block over Greenland. Preece et al. (2022) pointed out different pathways to





blocking over Greenland in summer. And more recently, Teubler et al. (2023) found two modes of year-round variability in the dynamics of GL by applying EOF analysis and *k*-means clustering to low-pass filtered upper-tropospheric PV anomalies in the days before GL onset, linked to the occurrence of negative PV anomalies upstream and downstream of Greenland.

Using the quasi-Lagrangian PV perspective, we identify the negative PV anomalies (PVAs⁻) linked to blocking over Greenland and investigate their propagation and origin. This allows us to quantify if there exists a direct link between the PV anomalies upstream and downstream and blocking over Greenland. For each GL regime life cycle, we define the PVA⁻ that exhibits the highest spatial overlap with the GL regime mask (cf. Figure 1b, white contour) in the period +/- one day around GL onset as the 'onset PVA⁻'. The frequency of onset PVAs⁻ reveals that PVAs⁻ linked to blocking over Greenland originate

from both, upstream and downstream of Greenland, and thus indicate two pathways of PVAs⁻ to Greenland (grey shading in lower row of Figure 3). For the objective classification of the two pathways, an objective partitioning of the onset PVAs⁻ is performed by defining two areas around Greenland east and west of 52.5 °W (center of mass longitude in the year-round VAPV'-weighted composite, Figure 1b), and by determining in which of the two areas the center of mass location of an onset PVAs⁻ is more frequently found in the three days before GL onset. Eastern ones are assigned to the 'retrogression pathway',

western ones to the 'upstream pathway'. For the subsequent discussion of the pathways, Figure 3 shows the frequency of onset PVAs⁻ for the two pathways separately for selected time steps around the onset, and Figure 4 reveals the mean track of PVAs⁻ for the pathways.

        Upstream PVAs⁻ are located over the northern United States six days before the onset (Figure 3f) and propagate north-eastward towards Greenland (Figure 3g–j). Thereby, the PVAs⁻ exhibit a quick northward movement once they reached the

East Coast of the United States (Figure 4b). In contrast, PVAs⁻ belonging to the retrogression pathway are characterized by a northwestward propagation against the mean flow from northern Europe towards Greenland in the four days before GL onset (Figure 3f–j). However, if the PVAs⁻ are traced further back in time, we find their origin in the storm track over the central North Atlantic (Figure 4a). Even nine days before the onset, the anomalies are located upstream of the Greenland blocking region, revealing an eastward propagation of PVAs⁻ assigned to the retrogression pathway before they become stationary for

a short time over Europe. Because of the initial eastward movement of PVAs⁻ linked to the retrogression pathway, the assignment to the pathways depends on the chosen time period which is illustrated in Figure A3. The longer the selected period before the onset, the more onset PVAs⁻ are assigned to the upstream pathway. However, the differences in the assignments vary slowly and the chosen period of three days before the onset is sufficient, as this is the time period with the largest differences in the propagation of onset PVAs⁻.

From a year-round perspective, more GL life cycles are assigned to the retrogression pathway (58 %) than to the upstream pathway (42 %). This matches well with previous studies that point to blocking over northern Europe (and in particular Scandinavia) as a precursor pattern of GL (Vautard, 1990; Michel and Rivière, 2011; Büeler et al., 2021). Interestingly, in Teubler et al. (2023) less life cycles are accounted to their retrograde cluster (50 %) and more to their upstream cluster (50 %), which can be explained by the different classification techniques between Teubler et al. (2023) and our study, i.e. low-pass filtered,

time-averaged EOF/*k*-means clustering vs. instantaneous PV fields for quasi-Lagrangian tracking. The seasonal stratification





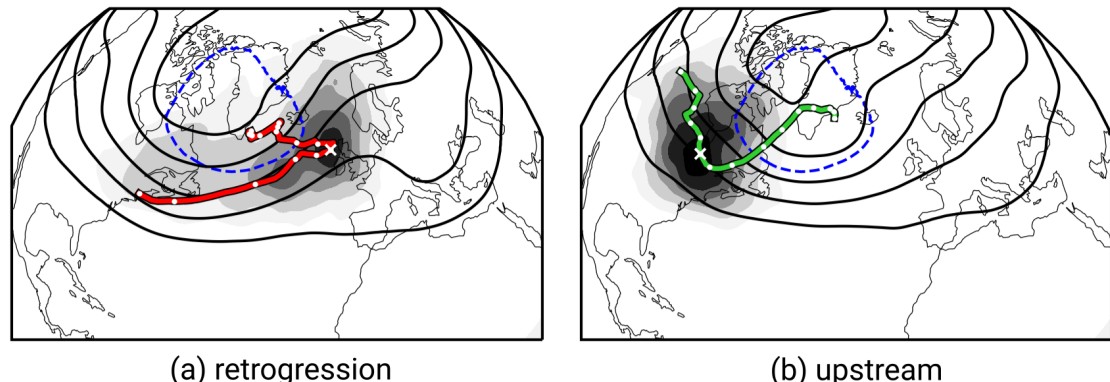

**Figure 4.** Mean track of onset PVAs⁻ for (a) the retrogression pathway and (b) the upstream pathway. The tracks are constructed by local maxima in onset PVA⁻ frequency for each time step (+/- 12 hours) for the period -9 days to +5 days around the GL onset and are shown in red (green) for the retrogression (upstream) pathway. Grey shading show the mean onset PVA⁻ frequency four days before the GL onset (t = -96 hours). For smoother tracks, a rolling mean window of +/- 12 hours was applied to the mean latitude and longitude points of the track. The white cross marks the time t = -96 hours and white points label the mean position of onset PVAs⁻ (along the track) in a temporal distance of 1 day. Black contour lines show the Z500 composite at the time t = -96 hours in steps of 52000, 53000, 54000, 55000, 56000 gpm. The blue dashed contour marks the regime mask of GL.

**Table 1.** Number of GL life cycles that are associated with the two pathways of PVAs⁻ to Greenland prior to GL onset. Percentages in brackets indicate the percentage of LCs that fall into the pathways for the year-round column (2nd column), while the other percentages point to changes in the relative share of LCs assigned to the pathways (in %).

|  | year-round | NDJFM | MJJAS | DJF | MAM | JJA | SON |
|---|---|---|---|---|---|---|---|
| all | 177 | 58 | 81 | 31 | 58 | 52 | 36 |
| retrogression | 102 (58 %) | 38 (+8 %) | 46 (-1 %) | 24 (+19 %) | 34 (+1 %) | 27 (-6 %) | 17 (-11 %) |
| upstream | 75 (42 %) | 20 (-8 %) | 35 (+1 %) | 7 (-19 %), | 24 (-1 %) | 25 (+6 %) | 19 (+11 %) |

(Table 1) reveals differences between seasons, with a dominance of the retrogression pathway in winter (77 %) but slightly more onset PVAs⁻ following the upstream pathway in autumn (53 %).

### 3.3 Amplitude evolution of onset PVAs⁻

We apply the PV anomaly amplitude metric introduced in Section 2.3.2 to all GL onset PVAs⁻. The following PV tendencies in Equation 2 are integrated on selected isentropic surfaces over the area of onset PVAs⁻: UP, LOW, $DIV_{adv}$, $DIV_{div}$, RES and $\mathcal{B}$nd. This reveals the contributions of different processes to the amplitude evolution, and, more importantly, shed light on the importance of dry and moist processes.



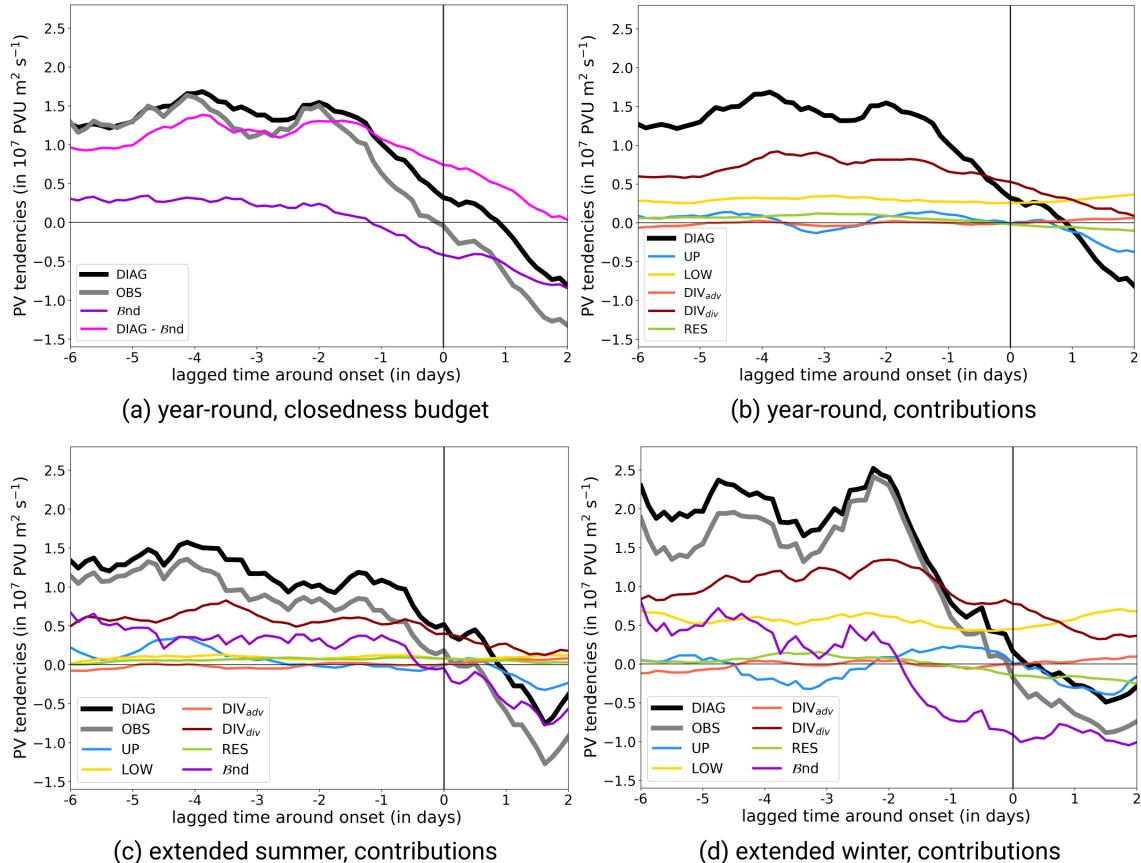

**Figure 5.** Mean amplitude evolution of onset PVAs$^-$ around GL onset. (a) Diagnosed (DIAG, black) and observed (OBS, gray) change of the onset PVA$^-$ amplitude. The boundary term $\mathcal{B}$nd (dark violet) and the difference of DIAG - $\mathcal{B}$nd (pink) is shown in colored lines. (b) Contribution of amplitude-modifying processes to the full diagnosed change in amplitude (DIAG): upper-tropospheric wave dynamics (UP, -$\mathbf{v}_{up} \cdot \nabla q_0$, blue), baroclinic interaction (LOW, -$\mathbf{v}_{up} \cdot \nabla q_0$, gold), advection part of the divergent outflow term (DIV$_{adv}$, -$\mathbf{v}_{div} \cdot \nabla q_0$, light red), divergence part of the divergent outflow term (DIV$_{div}$, $q'(\nabla \cdot \mathbf{v}_{div})$, dark red), and the residual PV tendency term (RES, -$\mathbf{v}_{res} \cdot \nabla q_0$, yellow green). Positive means a contribution to the strengthening of the amplitude, negative a contribution to the weakening of the amplitude. (c) Same as (b) but for GL regime LCs in extended summer (May–September). (d) Same as for (b) but for GL regime life cycles in extended winter (November–March). Note that all curves are smoothed by taking into account the time steps +/- 12 hours around.

**Year-round perspective**

Figure 5a–b provides first year-round insights into the amplitude evolution of onset PVAs$^-$ independent of the pathway and the PV tendency terms that contribute to the change in PVA$^-$ amplitude. Ideally, the diagnosed amplitude change DIAG (black line, DIAG = UP + LOW + DIV$_{adv}$ + DIV$_{div}$ + RES + $\mathcal{B}$nd) should represent the observed amplitude change OBS (grey line), measured as forward difference in the integrated PVA$^-$ amplitude. Based on the listed reasons for deviations (cf. Section B), we have filtered out questionable time steps when OBS and DIAG exhibit very different values. Using the condition |OBS



– DIAG| $< 2.5 \ 10^7$ PVU m$^2$ s$^{-1}$ ensures that time steps when DIAG deviates very strongly from OBS are excluded, but at
the same time a large fraction of values are still included in the composite. Still, DIAG somewhat over-estimate the actual
amplitude change (Figure 5a), but the temporal variations of the curves are very similar and the agreement is thus sufficiently
good for our analysis.

From a year-round perspective and independent of the pathway, onset PVAs$^-$ continuously undergo amplification in the days
before the onset (Figure 5a). The major contribution to the amplification arises from DIV$_{div}$ and indicates the importance of
moist processes for the development of onset PVAs$^-$ (Figure 5b). This agrees well with previous studies, which conclude that
rapid amplification of ridges is often strongly related to upper-level divergent outflow linked to mid-tropospheric latent heat
release in midlatitudes (e.g., Davis et al., 1993; Grams et al., 2011, 2018; Teubler and Riemer, 2021). A further contribution to
the amplification of onset PVAs$^-$ arises from LOW. This points to a suitable phase shift of the upper-tropospheric wave with the
lower-tropospheric temperature wave, hence leading to baroclinic amplification. The near constant amplifying contribution of
LOW has also been found for ridges within RWPs in the study of Teubler and Riemer (2021). $\mathcal{B}$nd shows a positive contribution
to the amplification (Figure 5a). Thereby, the sub term describing the change in area (last term on right hand side of Equation
B1) is throughout positive before the onset and indicates a growth in PVA$^-$ size towards the onset (not shown). In contrast,
the part of $\mathcal{B}$nd describing the divergence of the PV anomaly flux is negative, suggesting that low-PV air is advected out of
the region of the PVA$^-$ (not shown). The contributions of UP, RES and DIV$_{adv}$ are of minor importance for the amplitude
evolution and are just shown for consistency (Figure 5b).

The amplification of the PVA$^-$ amplitude starts to decrease in the two days before the onset, and, at the time of the onset, it
turns negative, indicating a beginning weakening of the amplitude (Figure 5b). The decrease in amplitude starting around the
GL onset is mostly driven by (i) a decreasing but still positive contribution of DIV$_{div}$, (ii) negative contributions of UP, and
(iii) a decrease in $\mathcal{B}$nd (Figure 5a).

Comparing these results with the Eulerian perspective on GL by Teubler et al. (2023), distinct differences exist in the
relative contributions of dry and moist dynamics. Teubler et al. (2023) diagnosed a dominance of the dry-dynamical linear
quasi-barotropic dynamics and (nonlinear) eddy flux convergence in the local buildup of the GL regime, with only small and
minor contributions linked to divergent PV tendencies and therefore moist processes (their Figure 7d). As discussed in detail
in Hauser et al. (2023b), these discrepancies emerge as a result of different perspectives and metrics used for a budget analysis.
The key role of divergent PV tendencies elaborated from the quasi-Lagrangian perspective is missed from the Eulerian point
of view, as onset PVAs$^-$ do not develop in-situ over Greenland. As a result, the moist processes occurring remotely from the
regime region are not taken into account from a Eulerian perspective.

**Seasonal stratification**

Previous studies investigated the seasonality in the PV tendencies and therefore in the PV dynamics (Teubler and Riemer,
2021). The lower row in Figure 5 shows the amplitude evolution and PV tendency contributions for extended summer (May–
August) and extended winter (November–March) separately.





Onset PVAs$^-$ experience a much stronger amplification during winter compared to summer before GL onset (Figure 5c,d). A close look into the contributions reveals seasonal differences in the strength of PV tendencies. Most prominently, LOW exhibits stronger positive contributions in winter. This has also been discussed in Teubler and Riemer (2021, their Figure 7)

and is the result of generally stronger baroclinicity in winter, leading to larger contributions by the baroclinic interaction term LOW. DIV$_{div}$ shows stronger values in winter, most probably due to the more frequent occurrence of WCBs in winter (e.g., Madonna et al., 2014). In total, it is the lack of baroclinic coupling and the weaker upper-level divergence that leads to a less strong amplification of onset PVAs$^-$ before GL onset in summer (Figure 5c,d).

**Breakdown into the two pathways**

Two different pathways of onset PVAs$^-$ to Greenland were revealed in Section 3.2 in the days before the onset. In the following, we investigate the year-round dynamics of onset PVAs$^-$ in dependence on the pathway. Figure 6 shows the amplitude evolution of onset PVAs$^-$ and the respective contributions. Spatial composite maps of the different PV tendency contributions are shown in Figure 7 and reveal the spatial pattern relative to the respective longitude-latitude center of mass position of onset PVAs$^-$.

The amplitude evolution of onset PVAs$^-$ shows distinct differences depending on the pathway before the GL onset (Figure

6a,e). The maximum amplification for onset PVAs$^-$ following the retrogression pathway takes place around five days before, while onset PVAs$^-$ from upstream experience their largest amplification shortly before the onset (1–2 days before). After the onset, the amplitude change matches well between the pathways. The diversity of the amplitude evolution raises the question whether the relative contribution of the PV tendency terms differs between the pathways.

The early peak in amplification for retrograding PVAs$^-$ is dominated by high contributions of DIV$_{div}$ (Figure 6a,f), which

amplify the onset PVAs$^-$ in particular on the northwestern edge of the anomaly (Figure 7a). Moist processes play a dominant role for the amplification at this early stage, when onset PVAs$^-$ are located over the eastern North Atlantic (cf. Figure 3f,g). At the same time, the high positive contribution of $\mathcal{B}$nd is purely attributable to a large growth in the PVA$^-$ area (not shown), and suggest that divergent outflow in the upper troposphere leads to the amplification and, in particular, to an increase in ridge area (e.g., Grams et al., 2018). UP contributes additionally to an amplification, pointing to an asymmetry of the positive

PV anomalies that flank the onset PVA$^-$. And indeed, the negative tendencies of UP on the upstream flank dominate over the positive tendencies of UP downstream (Figure 7a), which leads to an amplification upstream and the observed westward propagation of the PVAs$^-$. Upstream PVAs$^-$ show a local minimum in amplification when retrograding onset PVA$^-$ are strongly amplified a few days before GL onset (Figure 6a,e). A comparison between the contributions for each pathway reveals a lower contribution of LOW, DIV$_{div}$ and $\mathcal{B}$nd, which points to a less baroclinic-driven development (Figure 6c,f,h).

Upstream PVAs$^-$ experience their strongest amplification in the three days before GL onset, when the PVAs$^-$ reach the East Coast of the United States and head northwards towards Greenland (Figure 6a, Figure 4b). The amplification is dominated by contributions of DIV$_{div}$ and $\mathcal{B}$nd, similar as for the early peak of retrograding PVAs$^-$ (Figure 6f,h). The strong amplifying tendencies of DIV$_{div}$ are found in the northwestern corner of the onset PVAs$^-$ (Figure 7f,g,h). The positive contribution of $\mathcal{B}$nd points to an increase in the PVA$^-$ area at that time, which, again, could be linked to the divergent outflow in the upper

troposphere. LOW shows additionally a local maximum pointing to baroclinic coupling of the upper-tropospheric wave with the





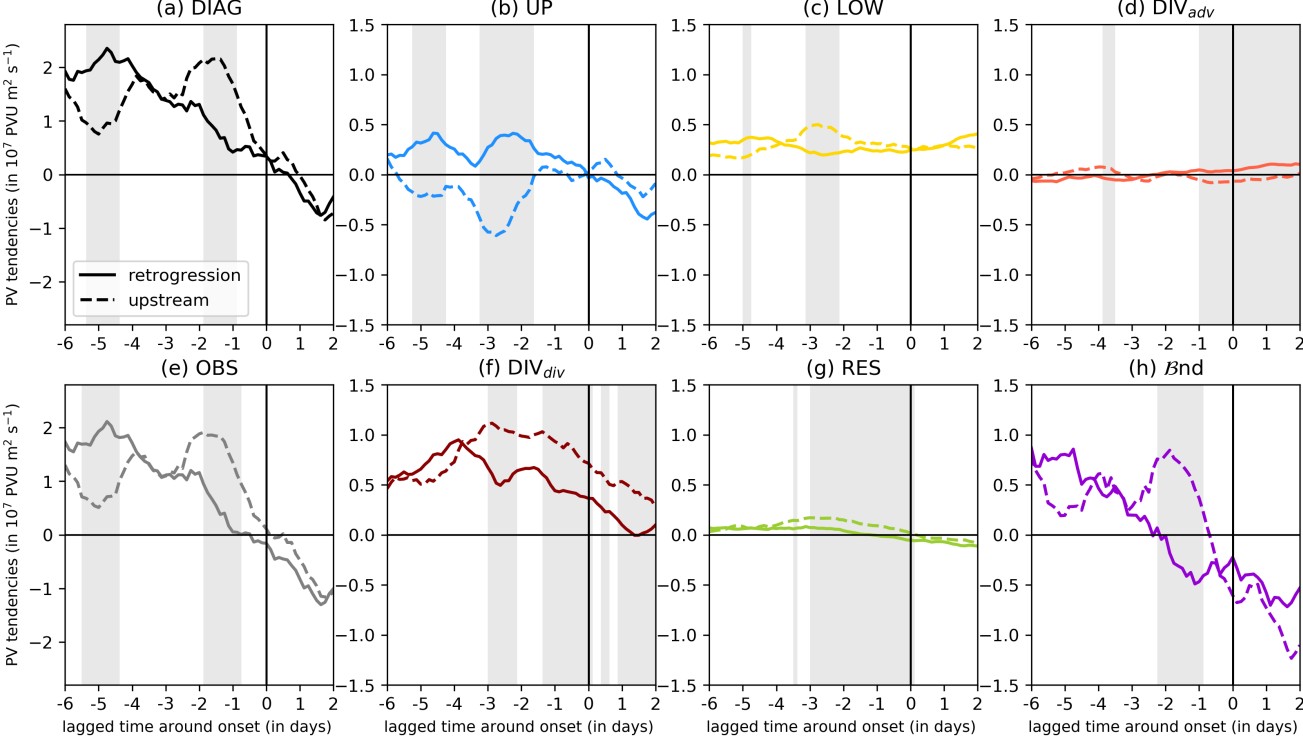

**Figure 6.** Mean amplitude evolution of onset PVAs$^-$ around GL onset following the upstream (dashed lines) or retrogression pathway (solid lines): (a) Diagnosed amplitude change (DIAG), (b) upper-tropospheric wave dynamics (UP), (d) baroclinic interaction (LOW), (d) advection by divergent wind field (DIV$_{adv}$), (e) observed amplitude change (OBS), (f) divergence of divergent wind field (DIV$_{div}$), (g) residual PV tendency term (RES), (h) boundary term ($\mathcal{B}$nd). Positive means a strengthening of the amplitude, negative a weakening of the amplitude. Grey shading indicates when the dynamics of the two pathways are significantly different from each other (re-sampling, Monte Carlo, 10.000 iterations, 2/98 %). Note that all curves are smoothed by taking into account the time steps +/- 12 hours around.

surface temperature wave, which adds to the amplification of the existing onset PVA$^-$ (Figure 6c, Figure 7f,g). Interestingly, UP counteracts the amplification in the days before the GL onset, which contrasts with the processes leading to maximum amplification of retrograding PVAs$^-$. This is linked to a trough that extends downstream of the PVA$^-$ and advects high PV from the north into the PVA$^-$ on its eastern flank (Figure 6b, Figure 7f,g). The large negative contribution of UP dampens the amplification of PVAs$^-$ from upstream significantly in the days before the onset.

**Summary of onset PVA$^-$ amplitude evolution**

The analysis on the amplitude evolution of onset PVAs$^-$ reveals a strengthening of the onset PVA$^-$ amplitude in the days before GL onset. We found that the temporal evolution and strength of amplification exhibits larger differences between the two pathways than between seasons (summer vs. winter). This is in good agreement with the Eulerian perspective of Teubler



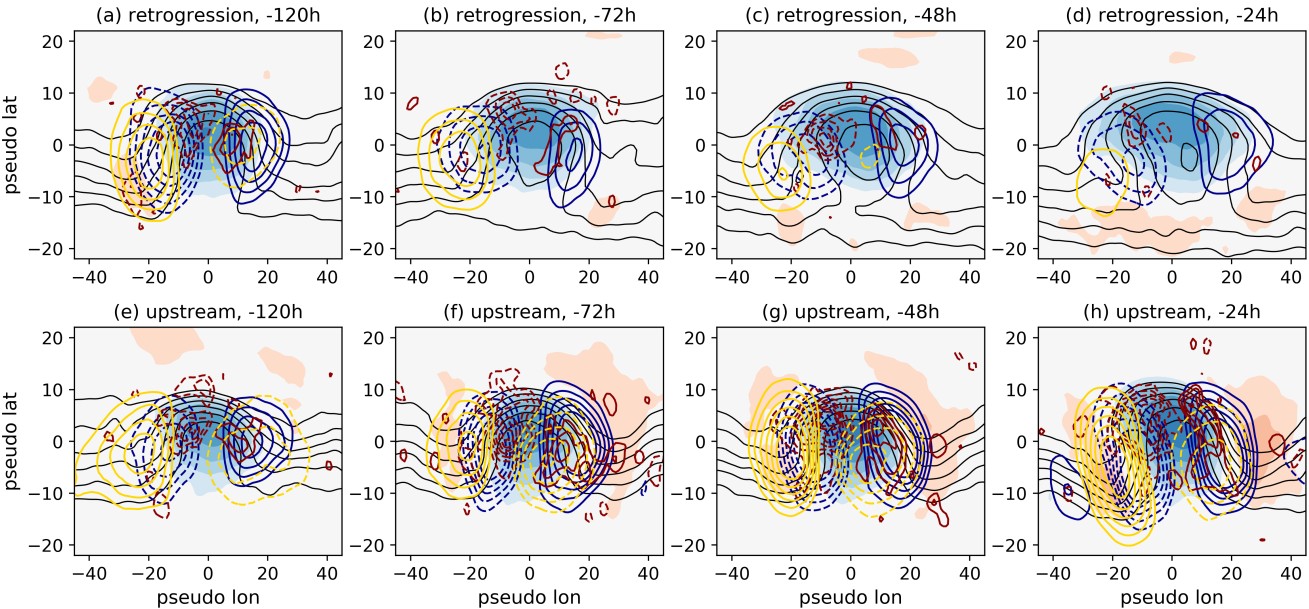

**Figure 7.** Centered composites on onset PVAs$^-$ (center of mass positions) for different time steps relative to the onset date (columns) and the two pathways (rows). Shading shows upper-tropospheric PV anomalies (in PVU) and black contours show the absolute PV in the upper troposphere (for the levels 2.5 to 5 PVU in steps of 0.5 PVU). The main contributing three PV tendency terms are shown in colored contour lines (i) UP (blue) in steps of +/- 10, 14, 18, 22, 26 · $10^7$ PVU m$^2$ s$^{-1}$, (ii) LOW (gold) in steps of +/- 1.2, 1.6, 2, 2.4, 2.8, 3.2 · $10^7$ PVU m$^2$ s$^{-1}$, and (iii) DIV$_{div}$ (red) in steps of +/- 3, 5, 7, 9, 11, 13, 15 · $10^7$ PVU m$^2$ s$^{-1}$. Solid and dashed contours mark positive and negative PV tendencies, respectively. The tendencies of DIV$_{div}$ and the absolute PV field were smoothed with a Gaussian filter ($\sigma = 1$).

et al. (2023), who found larger variation in the dynamics between their two EOF and clustering-based pathways compared to a pure seasonal stratification. In contrast to Teubler et al. (2023), the quasi-Lagrangian perspective quantifies an important and dominant role of moist processes in the formation and in particular in the amplification of PVAs$^-$ linked to blocking. As discussed in detail above and in Hauser et al. (2023b), this is because moist processes occur mostly off the blocked region and thus appear only as weak contribution in Eulerian frameworks focusing on the local circulation pattern. This also illustrates the importance of a multi-faceted view on blocking dynamics.

## 3.4 Link of DIV$_{div}$ tendency to moist-dynamical WCBs

The previous section highlighted the dominant contribution of DIV$_{div}$ to the amplification of onset PVAs$^-$ for both pathways. Many studies have linked divergent outflow close to the tropopause to moist processes below, and in particular to the presence of WCBs. Hence, DIV$_{div}$ has often been referred to as an indirect moist contribution. We here investigate in detail the link of the evolution of DIV$_{div}$ (Figure 6f) and the occurrence of WCBs in the immediate vicinity of the onset PVAs$^-$ (Figure 8) for the two pathways.





Onset PVAs⁻ linked to GL onset are often strongly amplified by $\mathrm{DIV}_{div}$ once they are located over the North Atlantic (Figure 6f, Figure 4). This suggests a link to WCB activity, as the climatological WCB activity exhibits a local frequency maximum over the storm track region in the North Atlantic (cf. Madonna et al., 2014). Retrograding PVAs⁻ experience the

maximum contribution of $\mathrm{DIV}_{div}$ to the amplification around four days before the onset over the eastern North Atlantic (Figure 6f, Figure 4a). In comparison, onset PVAs⁻ following the upstream pathway experience a decreasing contribution of $\mathrm{DIV}_{div}$ to the amplification once they reach the East Coast of the United States. The different timing in reaching the North Atlantic thus provide explanations for the variability of the DIV contribution to the onset PVA⁻ amplitude evolution.

Centered composites of upper-tropospheric PV anomalies and WCB activity on the onset PVAs⁻ (Figure 8) highlight WCB

activity in the upstream flank of the PVAs⁻, which is in good agreement with the theoretical understanding that WCBs amplify a downstream ridge in the upper troposphere (e.g., Wernli, 1997; Grams et al., 2011) and (cf. Figure 1 of Quinting and Grams, 2021). WCB inflow in the lower troposphere is located to the southwest of the upper-tropospheric PV⁻, WCB ascent is slightly shifted to the north of WCB inflow and in the southwestern corner of the PVA⁻, and WCB outflow resides in the northwestern or even northern part of the PVA⁻.

A good spatial agreement is found for WCB outflow frequency and the occurrence of amplifying $\mathrm{DIV}_{div}$ contributions for both onset PVA⁻ pathways (Figure 8). Following the retrogression pathway, WCB outflow occurs consistently before and even shortly after onset (Figure 8, upper row). Although the contribution of $\mathrm{DIV}_{div}$ decreases in the four days before the GL onset in an integrated sense, it still strengthens the onset PVA⁻ amplitude around the onset (Figure 6f). The strongest WCB activity in the retrogression pathway occurs early before the onset (Figure 8a) and is most likely linked to synoptic-scale cyclone activity

upstream (cf. Figure 3, low MSLP in blue). The ongoing WCB activity on the upstream flank of retrograding onset PVA⁻ suggests that moist processes on the northwestern flank of the onset PVAs⁻ importantly aid the retrograding propagation by continuously rebuilding a negative PVAs⁻ on its upstream flank. The spatial extent of onset PVAs⁻ measured by the extent of PV anomalies (dashed contours) indicates that retrograding PVAs⁻ exhibit a larger and in particular more zonally elongated area than the more compact upstream PVA⁻ – a requirement for circulation anomalies to retrograde in the typical thinking

of barotropic wave propagation. However, the high frequency of WCB outflow identified here potentially extends the area of the onset PVAs⁻ on its western flank which might critically support the westward propagation of the onset PVAs⁻. This highlights that the retrogression of blocks might not be purely barotropic and therefore linked to dry dynamics, but also involve an important moist-diabatic component.

In comparison to the retrograding onset PVAs⁻, an increase of WCB outflow frequency towards the onset is found for

PVAs⁻ that reach Greenland from upstream (Figure 8, lower row). This points to a more important role of moist processes in the few days before the onset compared to earlier times for the onset PVAs⁻ following the upstream pathway and is in line with the higher contribution of $\mathrm{DIV}_{div}$ to the amplification (Figure 6f). In particular one day before GL onset, the structure of the WCB is well represented and we find a very accurate agreement between amplifying tendencies of $\mathrm{DIV}_{div}$ and high WCB outflow frequencies (Figure 8g).





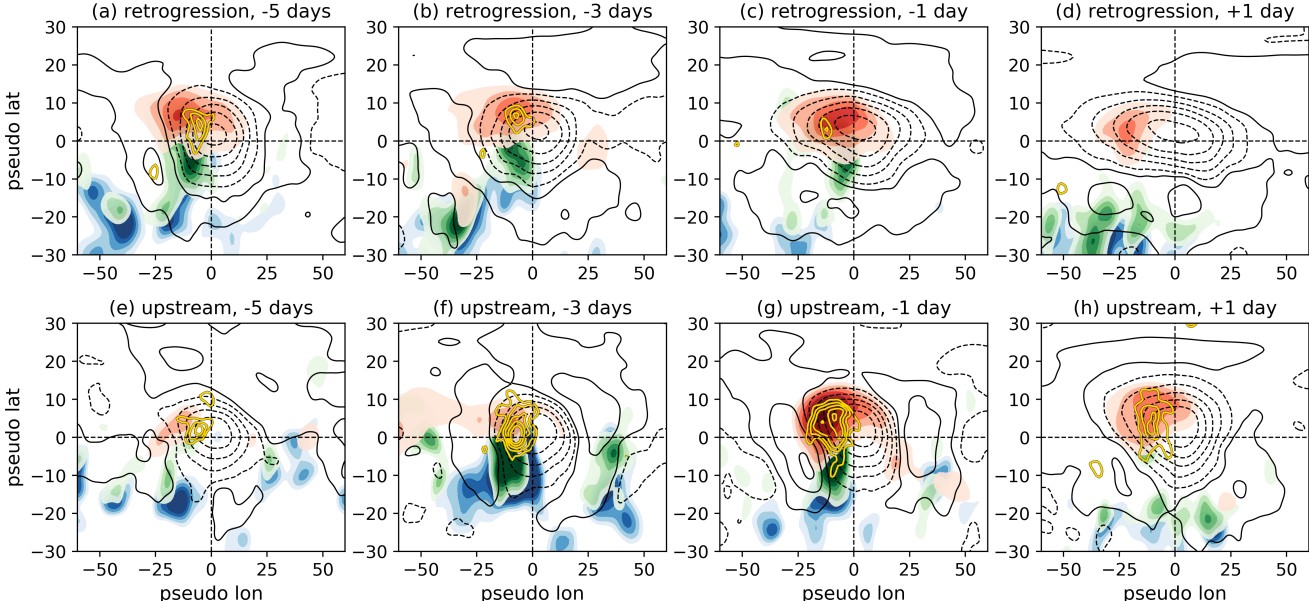

**Figure 8.** Composites centered on the position (center of mass) of onset PVAs$^-$ for selected times relative to GL onset (columns) and for the two pathways separately (rows) showing the occurrence of different WCB stages (shading) and VAPV' (black contours). Colored shading indicates the frequency of WCB inflow in the lower troposphere (blue, from 0.02 to 0.04 in steps of 0.005), WCB ascent in the mid-troposphere (green, from 0.02 to 0.045 in steps of 0.005), and WCB outflow in the upper troposphere (red, 0.06 to 0.16 in steps of 0.02). Black contours in solid and dashed illustrate the positive and negative VAPV' (500-–150 hPa), respectively. The contour levels displayed are [-1.3, -1.0, -0.7, -0.4, -0.1, 0.1, 0.4, and 0.7] PVU. PV tendencies of DIV$_{div}$ are shown in gold with contour levels of [-6, -8, -10, -12, -14] $10^6$ PVU m$^2$ s$^{-1}$. All fields shown are smoothed by a Gaussian filter with $\sigma = 2$.

Overall, the agreement between WCB outflow and the contribution of DIV$_{div}$ to the amplification of the onset PVA$^-$ amplitude matches well and therefore supports the close link between amplitude-strengthening DIV$_{div}$ tendencies and moist processes in WCBs.

## 4   Maintenance and decay of Greenland Blocking

Since the longevity of the GL pattern can provide conditions for extreme weather, it is of high importance to understand which
processes further strengthen GL after the onset until the maximum life cycle stage is reached. In addition, an understanding of the beginning decay phase after the maximum stage has been reached is still missing. Therefore, we investigate the PV dynamics around the maximum life cycle stage to provide insights into the key question: which processes maintain the block over Greenland and finally lead to the decay?





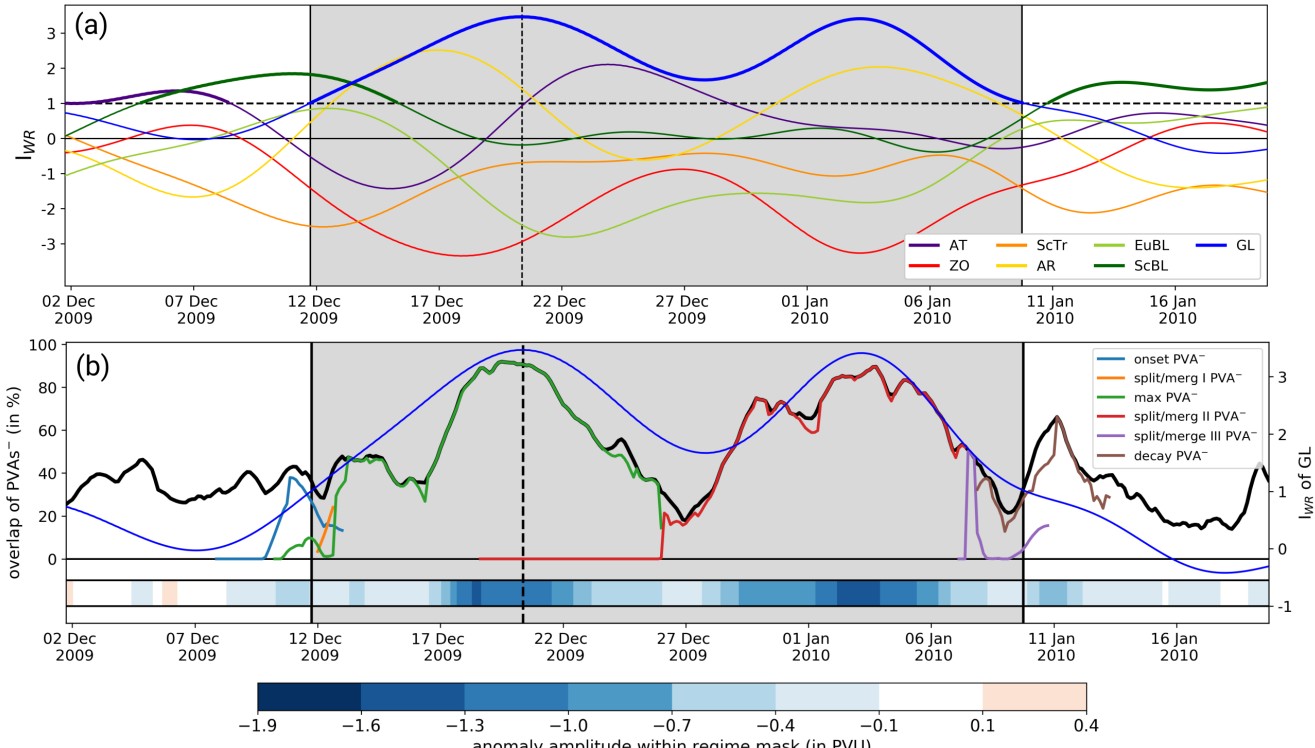

**Figure 9.** (a) Evolution of the $I_{WR}$ around the GL regime life cycle in winter 2009/2010. Each line shows the $I_{WR}$ for one of the seven weather regimes. Thick lines point to active regime life cycles (see Section 2.2 for details). Grey shading marks the lifetime of the GL regime life cycle from onset to decay. Vertical black lines point to the onset and decay time, dashed vertical line to the maximum stage. (b) $I_{WR}$ of GL (blue thin line, right y-axis), full overlap of PVAs$^-$ with the GL regime mask (black thick line, left y-axis), anomaly amplitude within the GL regime mask (horizontal bar in the lower part of the figure and colorbar). Each colored line (see legend) characterizes the temporal evolution of the overlap with the regime mask of a single PVA$^-$.

## 4.1 Contribution of several PVAs$^-$ to a full blocking life cycle

A complex behavior of PV anomalies during a blocking episode was previously noted in passing by Schneidereit et al. (2017) and indicated by Hauser et al. (2023b) for another blocking episode. To here demonstrate this complex behavior more explicitly, we briefly look at a long-lived GL regime life cycle in winter 2009/2010, which led to cold temperatures over Western and Northern Europe (e.g., Cattiaux et al., 2010).

The $I_{WR}$ of GL exhibits two maxima during the life cycle from 11 December 2009 to 10 January 2010 (Figure 9a), which is
in contrast to the unimodal course of the $I_{WR}$ for the European Blocking in March 2016 (see Figure 3 of Hauser et al., 2023b). This suggests phases of re-intensification of a blocked pattern for longer-lasting regime life cycles and enables a novel angle on blocking from a regime perspective, which allows for more transient behavior than classical blocking detection algorithms (e.g., Schwierz et al., 2004, with their strict overlap criterion). The temporal overlap of PVAs$^-$ with the regime mask over Greenland



closely reveals the same evolution as the $I_{WR}$ of GL (Figure 9b, black and blue lines). Several individual and independent

PVAs$^-$ dominate the anticyclonic circulation over Greenland intermittently, pointing to a transient behavior of PVAs$^-$ close to Greenland during an active GL life cycle (Figure 9b, colored lines). One PVA$^-$ exhibits a large spatial overlap with the regime mask and determines the onset of the GL life cycle (onset PVA$^-$, blue line in Figure 9b). However, within the regime life cycle, another PVA$^-$ propagates towards Greenland (not shown) and merges with the onset PVA$^-$ (green line). This PVA$^-$ solely describes the block over Greenland for the period 13–26 December. After the regime maximum, the $I_{WR}$ of GL decreases

temporarily, followed by another increase around 27 December (Figure 9a). The overlap of the PVA$^-$ around maximum stage with the regime mask decreases towards this local minimum in $I_{WR}$ (green line, Figure 9b), but then merges with a new PVA$^-$ (red line) around 26 December, which propagates towards Greenland. This new PVA$^-$ is linked to the second peak in $I_{WR}$ on 3 January 2010 (Figure 9a), which constitutes a reinforcement of the GL regime life cycle. Although it undergoes splitting and merging events (red to violet to brown lines, Figure 9b), this PVA$^-$ remains over Greenland in the period from the second $I_{WR}$

peak to the end of the life cycle (9 January) and even beyond.

This illustrative case highlighted that different PVAs$^-$ represent the block over Greenland at different life cycle days, complicating a systematic investigation of blocking life cycles. An analysis was conducted on how many PVAs$^-$ contribute to a single GL life cycle, where only PVAs$^-$ were counted that exhibit a spatial overlap with the regime mask of at least 10 % for a minimum duration of 12 hours. Around 20 % of the GL life cycles were associated with a single PVA$^-$. Two PVAs$^-$ are linked

to a regime life cycle in 22 % of all regime life cycles, three PVAs$^-$ for 20 %, four PVAs$^-$ for 18 %, and more than four PVAs$^-$ were found in 20 % of all GL regime life cycles. The differences in the number of contributing PVAs$^-$ motivates the definition of a single maximum PVA$^-$ for each regime life cycle. Equivalent to the onset PVA$^-$, we define the maximum PVA$^-$ as the PVA$^-$, that has the largest spatial overlap with the GL regime mask in the period +/- 1 day around the GL maximum stage. We further refer to this PVA$^-$ as 'max PVA$^-$'. In the case of the GL regime life cycle in winter 2009/2010 (Figure 9b), the

PVA$^-$ represented by the green line would be identified as the max PVA$^-$. In 103 out of 177 GL life cycles (58.2 %), the max PVA$^-$ is the same PVA$^-$ as the onset PVA$^-$, which justifies the decision to define max PVAs$^-$ for the systematic analysis of life cycle dynamics around the maximum stage in the following.

## 4.2 Evolution of max PVAs$^-$ around the maximum stage

From a traditional blocking perspective, the blocking anticyclone exhibits a slower propagation and is nearly stationary during

its maximum manifestation (e.g., Steinfeld and Pfahl, 2019). Such a behavior is also expected from the regime life cycle perspective around the maximum stage. However, the strengthening (weakening) of the $I_{WR}$ before (after) the maximum stage can be associated with either a strengthening (weakening) of a quasi-stationary PVA$^-$ within the regime mask or the migration of a PVA$^-$ into (away from) the regime mask. Figures 10 and 11 present snapshots of the large-scale PV pattern around the maximum stage, PV tendency composites centered on max PVA$^-$ and the amplitude evolution of max PVA$^-$ including the

different contributions.





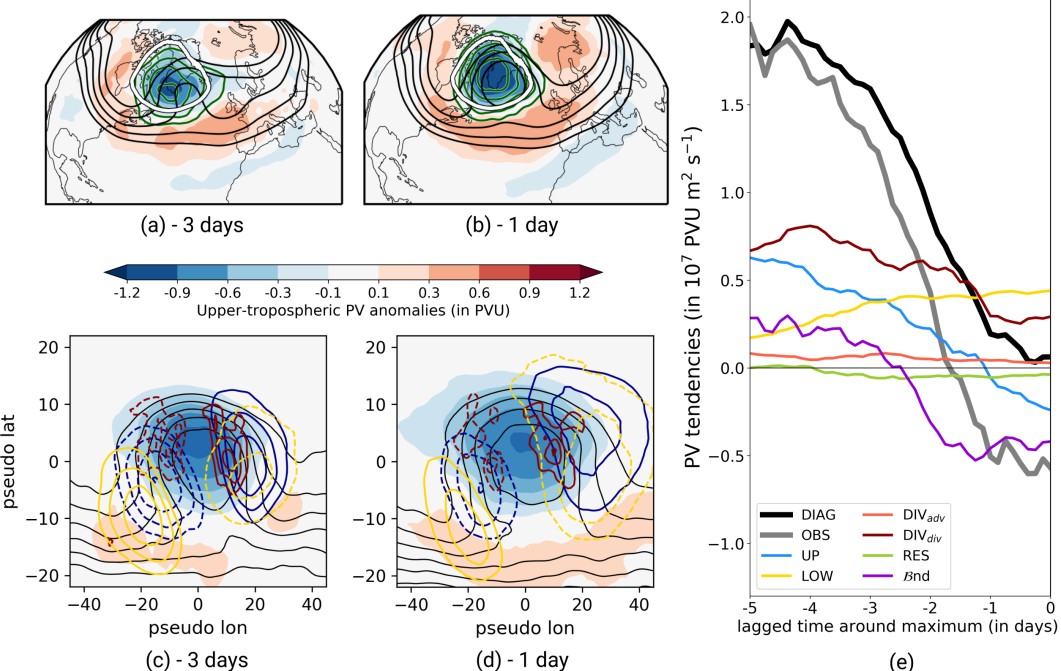

**Figure 10.** Year-round development of max PVAs$^-$ before the GL maximum stage. (a)-(b) Vertically-averaged PV anomalies (500–150 hPa, in shading), Z500 (black contour lines, ranging from 53500 to 55500 in steps of 500 gpm), and the occurrence frequency of max PVAs$^-$ (green contour lines ranging from 0.3 to 0.7) for selected time steps relative to GL maximum stage. The white thick contour shows the regime mask of GL. (c)-(d) Centered composites on max PVAs$^-$ for the same time steps as in (a) and (b). See the figure caption of Figure 7 for the explanations of the different contours. (e) Mean amplitude evolution of max PVAs$^-$ before GL maximum stage (grey line) and contributing processes (colored lines) making up the diagnosed amplitude change (black line). Note the applied temporal smoothing (+/- 12 hours).

**Evolution before the maximum stage**

In the days before the maximum stage, the GL pattern still sets in, and in particular the ridge over Greenland intensifies and spreads northward (Figure 10a,b). The troughs upstream and downstream of Greenland further deepen and create the U-shaped area of positive upper-tropospheric PV anomalies, which is evident in the mean composite of GL regime life cycles (cf. Figure 1b). A stationary behavior is identified for max PVAs$^-$ around the maximum stage within the GL regime mask (green contour lines in Figure 10a,b). This fits the findings of Steinfeld and Pfahl (2019), who looked at the maximum stage of blocking where a reduced propagation speed of blocks was found compared to the onset stage.

The stationarity of the max PVAs$^-$ and the increasing $I_{WR}$ towards the maximum stage suggest changes in the amplitude of the max PVAs$^-$. And indeed, we find that max PVAs$^-$ are further amplified in the days before the maximum stage (Figure 10e). The observed strength of the amplification declines towards the maximum stage and turns negative two days before the maximum stage (grey line), indicating a starting decrease in max PVA$^-$ amplitude.



The strengthening in the max PVA$^-$ amplitude before the maximum stage is dominated by DIV$_{div}$ (Figure 10e). The strong amplifying contributions of DIV$_{div}$ are located in the northwestern corner of max PVA$^-$, which is the typical location of moist-dynamical WCB activity linked to divergent outflow in the upper troposphere (Figure 10c,d). This points to an important contribution of moist processes to the reinforcement of blocks before they reach the maximum stage, and agrees well with investigations of previous studies (e.g., Steinfeld and Pfahl, 2019; Barrett et al., 2020). Additionally, UP makes a further and major contribution to the amplification of max PVAs$^-$. Strong amplifying tendencies of UP on the upstream flank are linked to the presence of a pronounced trough upstream of the max PVA$^-$ (Figure 10c,d,e). The contribution of UP declines simultaneously with the amplitude change and turns negative a day before the GL maximum stage. This is probably linked to the downstream development with the growing trough over Europe (Figure 10a,b). LOW exhibits a different behavior compared to DIV$_{div}$ and UP, with an increasing and overall positive contribution to the total amplitude change towards the maximum stage (Figure 10e). Negative tendencies of LOW prevail on the downstream flank of the max PVAs$^-$ and lead to strong amplification (Figure 10c,d). The decreasing contribution of UP and the increase in LOW suggest moist-baroclinic downstream development, as mentioned in Teubler and Riemer (2021). The positive contribution of $\mathcal{B}$nd in the period -5 to -3 days is dominated by the area change term (not shown) and indicates a natural growth of the max PVAs$^-$ or even a growth in anomaly size by merging of small-scale PVAs$^-$ into the max PVAs$^-$ (Figure 10e). In contrast, the negative contribution of $\mathcal{B}$nd shortly before the maximum stage indicates amplitude weakening, probably linked to a shrinking of the max PVA$^-$ size. This suggests that the starting decay of the regime after the maximum stage co-occurs with a decrease in amplitude of max PVAs$^-$.

**Evolution after the maximum stage**

The large-scale PV pattern exhibits similarities with the mean pattern of GL around one day after the maximum stage (Figure 11a), but already indicates a weaker ridge over Greenland compared to the day before the maximum stage (Figure 10b). Three days after the maximum stage (Figure 11b), the ridge further weakened and lost in northward extent. The upper-tropospheric PV anomaly field indicates an even retrograde behavior of the ridge towards Canada. From the perspective of max PVAs$^-$, the max PVA$^-$ frequency (green lines) remains nearly unchanged one day after the maximum compared to the day before the maximum (Figure 10b, 11a), indicating the stationarity of max PVAs$^-$ even in the days after the maximum stage. However, the frequency of max PVAs$^-$ declines and could indicate either a high variability in the position of max PVAs$^-$ or a local decay of max PVAs$^-$ over Greenland.

Max PVAs$^-$ experience a strong decrease in amplitude after the maximum stage (Figure 11e). The first contribution towards a decline in the amplitude is kicked off by $\mathcal{B}$nd (cf. Figure 10e), suggesting a weakening of max PVAs$^-$ by a decrease in anomaly area and eddy fluxes that advect low-PV air out of the max PVA$^-$ area. This is followed by UP, which switches the sign from positive to negative around one day before the maximum stage and exceeds the negative contribution of $\mathcal{B}$nd shortly after the maximum stage. UP as leading contribution to the decay of ridges has already been quantified in Teubler and Riemer (2021), was attributed to an asymmetry between the troughs upstream and downstream of the ridge, and essentially signifies downstream dispersion of Rossby waves. Here, the trough over Europe is more pronounced than the upstream trough, which leads to pronounced positive PV tendencies on the downstream flank of the max PVAs$^-$ (Figure 11a–d).





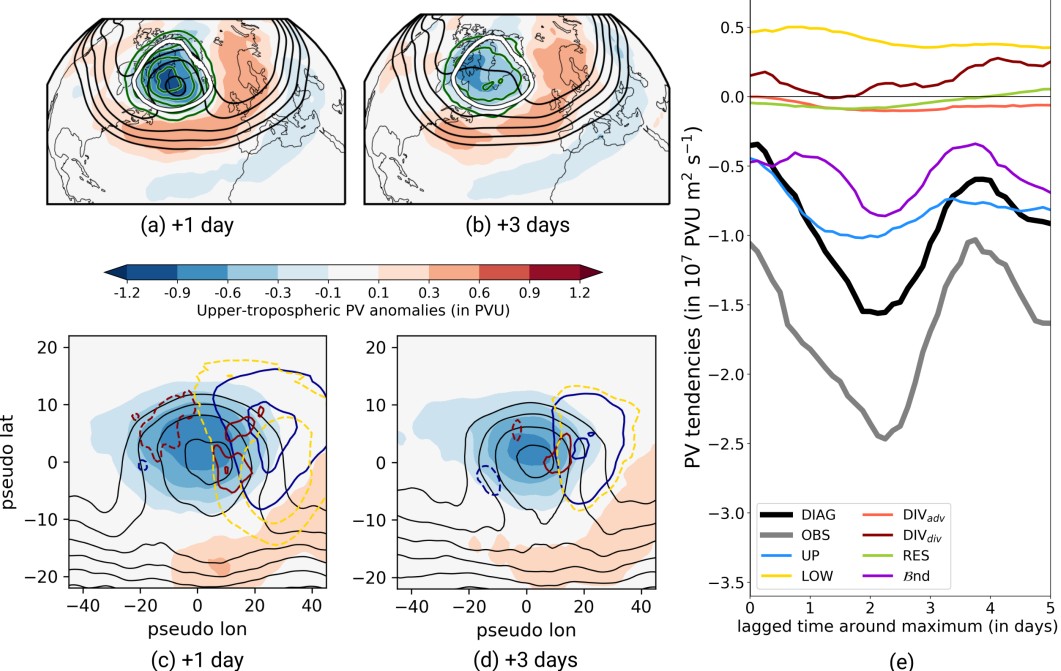

**Figure 11.** Same as Figure 10 but for the time period after the maximum stage.

The decrease in max $PVA^-$ amplitude is furthermore driven by a decreasing contribution of moist processes, as described by $DIV_{div}$ (Figure 11e). Figure 11c,d even shows dominating positive tendencies of $DIV_{div}$ southeast of the max $PVA^-$ center, probably linked to large-scale subsidence. This strong decrease in the contribution of moist processes to an amplification after the maximum stage agrees well with previous findings of Barrett et al. (2020), who found intense high integrated vapor transport (IVT) values before the maximum stage of Greenland Blocking and a strong decrease of IVT afterward. It furthermore supports the hypothesis of Hoskins (1997) that blocking decay is linked to the breakdown of the maintenance process, which is, in this case, the contribution of moist processes.

Whereas most PV tendency terms turn negative shortly before and after the maximum stage, LOW stays positive and counteracts the net decrease in amplitude (Figure 11e). This points to a favorable phase shift between the upper-tropospheric PV anomalies and the lower-tropospheric temperature wave, which leads to amplification of the max $PVAs^-$ by baroclinic coupling. Here, the strengthening takes place in the eastern half of the anomaly, i.e., on the downstream flank, and decreases with time lag after the maximum (Figure 11c,d).

### 4.3 Full life cycle dynamics

The length of GL regime life cycles is subject to high variability and can range from five to more than 21 days (Figure A1a,b). This limits the analysis in Section 4.2 that is performed for lagged days around the maximum stage. Therefore, in a last step, we want to account for the full life cycle from the onset to the decay stage. Figure 12 displays the mean net PV tendencies





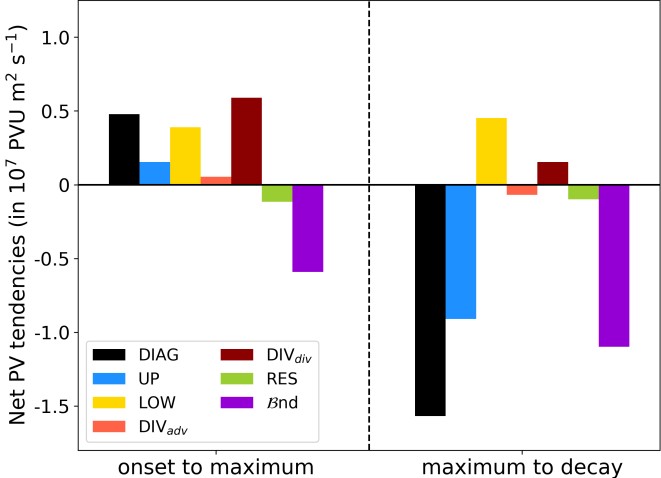

**Figure 12.** Net effect of amplitude evolution of max PVAs⁻ between the onset and maximum stage (left) and between the maximum stage and decay (right).

integrated over the max PVAs⁻ in the period from onset to decay for times, when the max PVA⁻ existed. Weighted by the total days between onset and maximum and maximum and decay, this allows for an investigation of the full life cycle, independent of life cycle length.

A positive net amplification of max PVAs⁻ occurs in the first part of the GL life cycle from onset to the maximum stage (Figure 12, left side). $\mathrm{DIV}_{div}$ is the key net contribution to the amplification, revealing a dominant importance of moist processes. The net contributions of LOW and UP are of secondary importance. $\mathcal{B}$nd dampens the amplification, although the net effects of the sub terms of $\mathcal{B}$nd (Equation B1) are of different sign, with a dominant dampening of the amplitude by the eddy flux convergence term but a slight strengthening as the results of a net growth in max PVA⁻ area. The net contributions of

$\mathrm{DIV}_{adv}$ and RES are negligibly small.

In contrast, max PVAs⁻ experience a net decrease in amplitude from the maximum to the decay stage of GL life cycles. The leading contributions to the decay arise from UP and $\mathcal{B}$nd. This time, both sub terms in $\mathcal{B}$nd weaken the amplitude, which points to a decrease in max PVA⁻ area (not shown). The net effect of LOW is strongly positive and highlights the baroclinic-coupling after the maximum stage. The strongly decreased contribution of $\mathrm{DIV}_{div}$ to the amplification of max PVAs⁻ compared to the

period before the maximum stage indicates the lack of moist processes, which would act as a maintenance mechanism for blocking over Greenland.

The insights obtained from the analysis of the full life cycle from onset to maximum and maximum to decay agrees well with the analyses in Sections 4.2 and 4.2, indicating that the investigation of the few days around the maximum stage could already be sufficient to disentangle the dynamics of the maximum stage. Furthermore, together with Figure 10, we can conclude that

the amplitude maximum of max PVAs⁻ is somehow connected to the maximum in the blocking regime life cycle.



## 5 Summary and conclusion

In this study, we systematically investigated the dynamics of Greenland Blocking based on ERA5 reanalysis data. For the first time we employed a novel quasi-Lagrangian PV perspective in a climatological way to disentangle contributions from dry and moist dynamical processes in a consistent framework. Using an objective blocking regime life cycle definition, insights were gained into the processes that govern the onset, maintenance and decay of blocking. A quasi-Lagrangian PV framework, originally developed in Hauser et al. (2023b), was applied to Greenland blocking life cycles to gain information on the propagation and origin of partly transient PVAs⁻ constituting the block over Greenland. Using a piecewise PV tendency framework, we were able to quantify the relative contributions of dry and moist processes in the amplitude evolution of PVAs⁻ for all Greenland blocking regime life cycles in 43 years of reanalysis.

Two distinct pathways of PVAs⁻ to Greenland were found in the days before Greenland blocking onset from the quasi-Lagrangian perspective. The first pathway ('upstream pathway') comprises PVAs⁻ that reach Greenland from the southwest and originated from North America. The second and dominating pathway ('retrogression pathway') describes the propagation of PVAs⁻ that originated also from North America, but are located over northern Europe a few days before the blocking onset and retrograde westward towards the onset time. Often, this retrogression has been linked to cyclonic Rossby wave breaking before Greenland blocking events (Woollings et al., 2008; Michel and Rivière, 2011), or to eddy forcing on the synoptic scale (Mullen, 1987; Teubler et al., 2023). Multiple studies pointed to blocking over Scandinavia as precursor pattern (e.g., Vautard, 1990; Büeler et al., 2021) or identified retrograding PV anomalies linked to blocking to the east of Greenland before blocking (e.g., Croci-Maspoli et al., 2007; Preece et al., 2022; Teubler et al., 2023). Here, we were able to systematically extract, for the first time, the different pathways of PVAs⁻ to Greenland from the novel quasi-Lagrangian approach developed in Hauser et al. (2023b).

The investigation of the amplitude evolution of PVAs⁻ on their way to Greenland showed a continuous amplification in the days before the blocking onset. Thereby, the timing in maximum amplification poses the main difference between the two identified pathways. Retrograding PVAs⁻ experience an early peak several days before the onset, when the PVAs⁻ still exhibit an eastward propagation from the North Atlantic region towards northern Europe. In contrast, upstream PVAs⁻ are strongly amplified later around 1–2 days before the onset. Although previous studies recommend splitting up process-based analyses into seasons, we found more distinct differences in the dynamics between the pathways than between individual seasons. This result is in line with Teubler et al. (2023), who obtained two different modes of variability to blocking over Greenland and showed that the differences in the dynamics are much more pronounced between the modes than between the seasons.

Divergent PV tendencies, which indirectly point to moist processes, play a key role in the amplification of PVAs⁻ independent of the pathway. This is in line with the studies of Pfahl et al. (2015) and Steinfeld and Pfahl (2019), who found an important contribution of diabatic heating to the formation of blocking. We were able to further link the increase in PVA⁻ amplitude by strong contributions of divergent PV tendencies to WCB activity over the North Atlantic. Although upstream PVAs⁻ are associated with more WCB activity, the quasi-Lagrangian perspective also revealed two periods of PVA⁻ amplification for PVAs⁻ following the retrogression pathway. First, retrograding PVAs⁻ originate from the western North Atlantic



early on before the GL onset and are amplified by moist processes during their propagation over the North Atlantic. Second, we show that the retrogression of PVAs⁻ not only occurs dry-dynamically, but is additionally enhanced by moist-dynamical activity on the anomalies' western flank, supporting the westward propagation towards Greenland. This confirms a previous hypothesis of Preece et al. (2022) on the additional role of diabatic processes for the westward displacement of blocks. The fact that moist processes are of primary importance for blocking onset from the quasi-Lagrangian perspective but are on minor importance from an Eulerian perspective in Teubler et al. (2023) are, at first glance, contradictory. However, in agreement with Hauser et al. (2023b), this is the result of different perspectives and rather complements the dynamical picture of blocking: Moist processes occur upstream or downstream of the Greenland blocking domain and are missed from the Eulerian approach that takes into account only processes within the blocking region. The propagation of PVAs⁻ from a quasi-Lagrangian point of view are contained in the strong dominance of the linear, quasi-barotropic dynamics and non-linear eddy fluxes of Teubler et al. (2023).

With a focus on the dynamics within the life cycle of blocking, we found that the amplitude evolution of the stationary PVAs⁻ is closely related to the maximum in the blocking life cycle. Main contributions to the further amplification of PVAs⁻ after blocking onset arise mainly from divergent PV tendencies linked to moist processes. This agrees well with previous results of Barrett et al. (2020), who found intense moisture fluxes along the western flank of particularly extreme Greenland blocking events, pointing to an important maintenance mechanism for blocks. Upper-level wave dynamics and baroclinic interaction play an amplifying role on the PVAs⁻ amplitude, too. This agrees also well with findings of Teubler and Riemer (2021), who studied mechanisms leading to the maximum amplitude in ridges as part of RWPs. After the maximum life cycle stage is reached, PVAs⁻ start to weaken. The contribution of moist processes declines, referring to an earlier point of Woollings et al. (2018) that the decay process of blocking is linked to the lack of a maintenance process. The main contribution to the amplitude decrease evolves from upper-tropospheric wave dynamics and from the boundary term, whereby the latter suggests a strong decrease in the PVA⁻ size and a diffusive effect of eddy fluxes within the PVA⁻ that announces blocking decay. This result was also reported in Teubler and Riemer (2021) for the weakening of ridges after they reached their maximum in amplitude.

In conclusion, we discovered two distinct pathways of PVAs⁻ for Greenland blocking using a novel quasi-Lagrangian perspective. In agreement with Pfahl et al. (2015) and Steinfeld and Pfahl (2019), the results emphasize the importance of moist processes in the formation and maintenance of blocking anticyclones, which motivates a close investigation of the involved processes in climate models and their role for predictability. We demonstrated that the moist-baroclinic evolution is hidden in terms describing Rossby wave dynamics and baroclinic interaction in Eulerian frameworks (e.g. Teubler et al., 2023). We are currently investigating the PV dynamics of other blocked regime types in the North Atlantic-European region from the quasi-Lagrangian perspective complementary to the Eulerian perspective by Teubler et al. (2023). Taken the two perspectives together supports a more complete understanding of blocking dynamics.

*Data availability.* The data are referenced in Section 2.1. ERA5 data are freely available at https://doi.org/10.24381/cds.bd0915c6 (Hersbach et al., 2020). The data for warm conveyor belt footprints in ERA5 reanalysis data by Quinting et al. (2022) are freely available at



https://gitlab.kit.edu/julian.quinting/elias-2.0. The codes and data from this study can be provided by the authors upon request. We plan to publish the quasi-Lagrangian tracking tool, and upper-tropospheric negative PV anomalies and the PV tendency fields for the ERA5 period
in the course of the review process.

*Author contributions.*  SH performed the analysis and wrote the paper. FT provided the piecewise PV tendencies. CMG provided the year-round North Atlantic–European weather regime data based on ERA5. FT, CMG, MR, and PK gave important guidance during the project and provided feedback on the paper.

*Competing interests.*  MR, PK and CMG are members of the editorial board of Weather and Climate Dynamics. The authors have no other
competing interests to declare.

*Acknowledgements.*  The research leading to these results has been done within the sub-project "Dynamics and predictability of blocked regimes in the Atlantic-European region (A8)" of the Transregional Collaborative Research Center SFB / TRR 165 "Waves to Weather" (www.wavestoweather.de) funded by the German Research Foundation (DFG). The contribution of CMG is funded by the Helmholtz Association as part of the Young Investigator Group "Sub-seasonal Predictability: Understanding the Role of Diabatic Outflow" (SPREADOUT,
grant VH-NG-1243). The authors would like to thank Dr. Stephanie Henderson and Dr. Jonathan Martin for discussions that emerged during a research visit at the University of Wisconsin-Madison. Further gratitude is expressed to the members of the "Large-Scale Dynamics and Predictability" working group at KIT for valuable discussions on this project.

## Appendix A:  Derivation of the piecewise PV tendency equation from a quasi-Lagrangian perspective

Equivalent to Teubler and Riemer (2016, 2021), we insert Equation 2 into the first term on the RHS of Equation 4 and use the
partitioning of the full wind field (cf. Equation 3). The second term on the RHS of Equation 4 has been directly calculated with the explicit calculation of $q_0$ in previous studies (e.g., Teubler and Riemer, 2016, 2021) and is negligible small compared to the first term. However, in agreement with Teubler et al. (2023) (see their appendix A), we translate Equation 2 into a tendency equation for $q_0$, such that we can take into account the contributions of different processes to the change in background PV:

$$\frac{\partial q_0}{\partial t} = < \frac{\partial q}{\partial t} > = < -\boldsymbol{v} \cdot \boldsymbol{\nabla}_\theta q > + < \mathcal{N} > . \tag{A1}$$

Here, <> is a mean operator that consists of averages between 1980–2019 for each calendar day and a subsequent running mean (+/- 15 days) – equivalent to the calculation of $q_0$ and $\mathbf{v}_0$. Putting together Equations 2,4 and A1, and using the partitioned wind field (Equation 3) yields the final equation for the amplitude evolution of PV anomalies in Equation 5. The main difference to Hauser et al. (2023b, their Equation 9) is the subtraction of a climatological background term for each sub term of the equation and the elimination of the term describing the advection of background PV by the background wind field ($-\mathbf{v}_0 \cdot \boldsymbol{\nabla} q_0$).





**Appendix B: Estimation of the boundary term $\mathcal{B}nd$ and reasons for deviations between DIAG and OBS**

The boundary term $\mathcal{B}$nd reveals similarities with $\mathcal{B}$nd in Hauser et al. (2023b) and can be estimated accordingly, as

$$
\begin{aligned}
\mathcal{B}nd = \oint_{S(t)} q'(\mathbf{v}_s - \mathbf{v})d\mathbf{S} &= -\int_{A(t)} \boldsymbol{\nabla} \cdot (\mathbf{v}q')\,dA - \int_{A(t)} < -\boldsymbol{\nabla} \cdot (\mathbf{v}q') > dA + \oint_{S(t)} q'(\mathbf{v}_s \cdot \mathbf{n})\,dS \\
&\approx -\int_{A(t)} \boldsymbol{\nabla} \cdot (\mathbf{v}q')\,dA - \int_{A(t)} < -\boldsymbol{\nabla} \cdot (\mathbf{v}q') > dA + \overline{q}' \cdot \Delta A,
\end{aligned}
\tag{B1}
$$

with $\overline{q}$ as the average of $q'$ along the boundary $S(t)$ and $\Delta \mathcal{A}$ taken from the observed area change of the PV anomaly. $\mathcal{B}$nd
exhibits major contributions to the amplitude change for (i) strong eddy flux divergence/convergence within the anomaly area, or (ii) when the area strongly increases/decreases between two consecutive time steps (occurs often during splitting and merging events).

The following factors (can) limit the closeness of the PV tendency budget as measured by the difference between DIAG and OBS: (i) the lack of NONCONS in this study, (ii) uncertainties from the partitioning of the wind fields, (iii) comparison
of instantaneous tendencies with a finite difference of three hours, (iv) limitation of PV tendencies to the domain 25–80 °N, and (v) abrupt changes in the PV anomaly area $A(t)$ that are often linked to splitting and merging and lead to exceptional high values of $\mathcal{B}$nd.




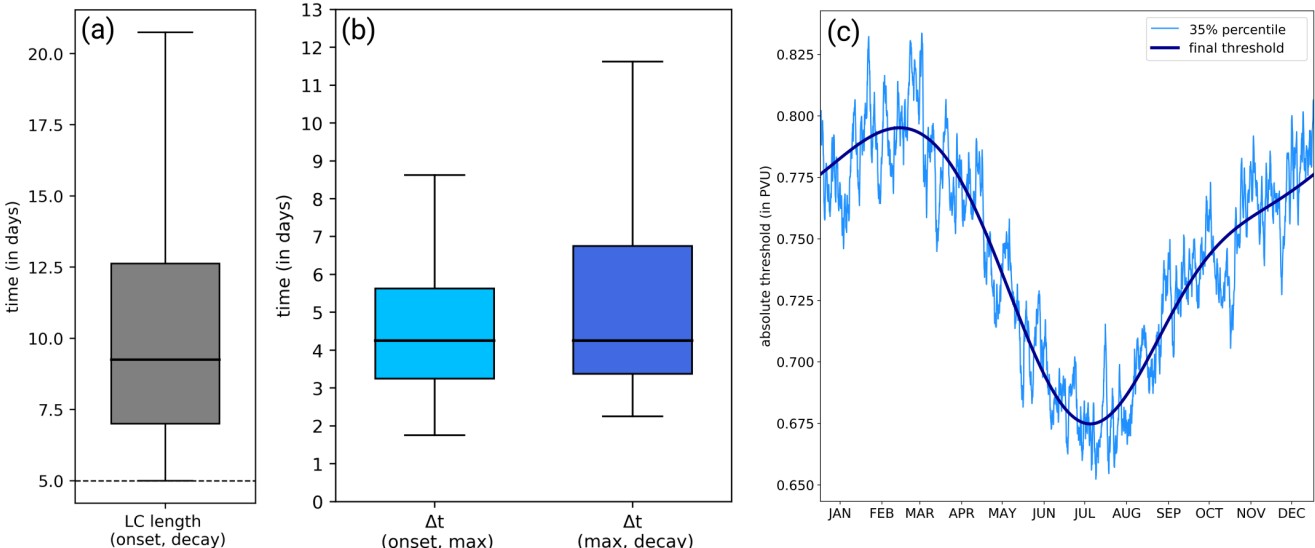

**Figure A1.** (a) Distribution of life cycle lengths (measured from onset to decay) in days for all year-round valid GL life cycles. (b) Distribution of the duration between the onset and maximum stage (left) and between the maximum and decay stage (right) in days. (c) Absolute threshold (in PVU) based on the running threshold analysis to define PVAs$^-$ (see Section 2.3.1 for details). The light blue line displays the 35 % of negative PV anomalies on the Northern Hemisphere for each day within the calendar year, and the dark blue line shows the smoothed curve based on fast Fourier transformation, which makes up the final running threshold used in this study.

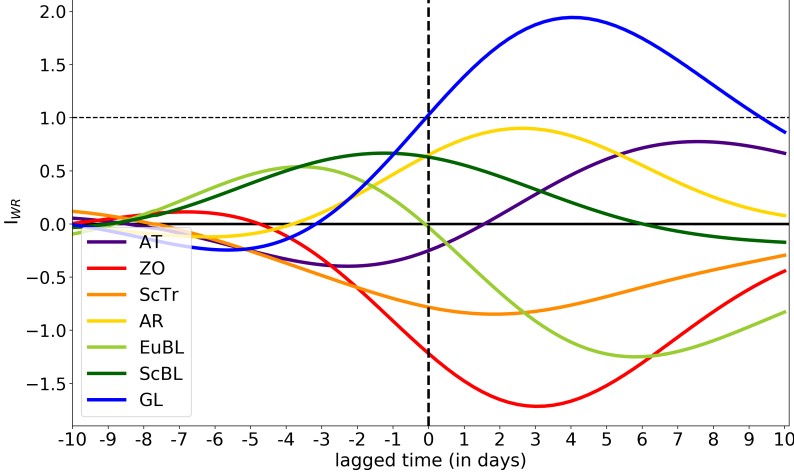

**Figure A2.** Lagged composite of the $I_{WR}$ for all seven weather regimes of Grams et al. (2017) around GL onsets in the ERA5 period 1979–2021.



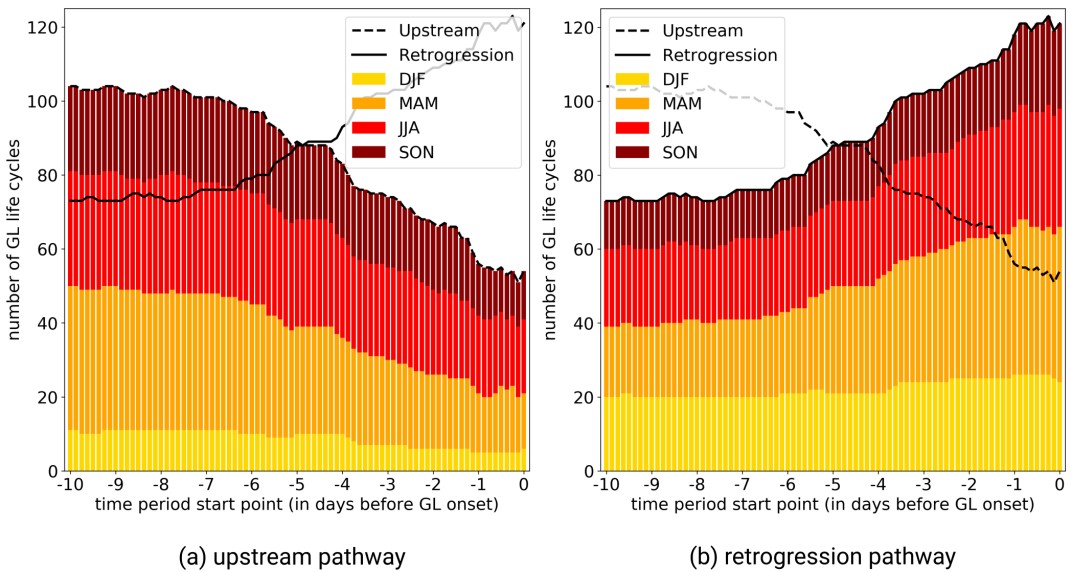

(a) upstream pathway          (b) retrogression pathway

**Figure A3.** Sensitivity of assignment of GL life cycles based on the period selected before GL onset for (a) the upstream pathway and (b) the retrogression pathway. The longer the selected period before onset, the more life cycles are assigned to the upstream pathway, which again illustrates that retrograde migrating PVAs$^-$ also have their original origin from upstream.

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
