# Peer review of "Life cycle dynamics of Greenland blocking from a potential vorticity perspective"

_EGUsphere, 2023_

## Author Response (AR1)

Author responses to the reviews on manuscript ID 2023-2945

**Life cycle dynamics of Greenland blocking from a potential vorticity perspective**

Hauser et al.

We thank all reviewers for their detailed feedback on the manuscript, which helped to improve not only the quality of the research, but also its presentation in the manuscript. Please find below the replies to the comments raised by all reviewers separately.
* * *
**Reviewer 1**
* * *
The evolution of Greenland blocks are examined here from a PV perspective. The authors show, by tracking negative PV anomalies reaching the Greenland region, that there are two distinct pathways through which Greenland blocking typically develops. The focus then is on quantifying the contribution of various terms in the PV tendency equation to the onset, maintenance and decay of Greenland blocks. It is shown that moist processes (via quantification of the divergence of the divergent wind within the PV anomaly) are a leading contributor to the onset and maintenance of Greenland blocks (with difference between each pathway quantified and explained). These findings add to the growing body of work on the diabatic contributions to block dynamics.

The paper is well written, the analysis techniques appropriate for the aims of the paper and the figures included are high quality and support the conclusions reached by the authors. In particular, I thought the introduction was well written and contains a nice introduction to the causes and impacts of atmospheric blocking in general (with specific results relating to Greenland nicely highlighted) and the quasi-Lagrangian PV perspective nicely allows the interpretation on the evolution of the PV anomalies during their entire lifecycle. It is also an interesting result that there are clearly two distinct pathways for negative PV anomalies to develop over the Greenland region, which opens many avenues for further study on the two different pathways and e.g. their representation in models.

The article fits well within the scope of WCD and includes many results that will be of interest to its readers. For the reasons listed above I believe the manuscript is suitable for publication in the journal subject to the authors addressing the minor comments listed below.

We would like to thank the reviewer for this very detailed and thorough and overall positive review.

**Minor comments:**

1. The authors use a weather-regime classification of Greenland blocking rather than a typical (feature-based) blocking index. What do the days that get assigned the GB weather regime look like? E.g. how many days go into the composite in Figure 1 and what is the corresponding spread among these days? Did you check that there are not days that get assigned to GB that a synoptician would not describe as blocked? As previous studies have shown results can be sensitive to the choice of blocking index (e.g. Barriopedro et al. 2010), it would increase the robustness of your results if they could be replicated using another index. I understand if this is not feasible but some discussion should be added to reflect this at least.

We thank the reviewer for this comment. First, the sample size of the composite is 15128 time steps (converted in days: 1891) given the used ERA5 period 1979-2021. Figure R1a shows the standard deviation of geopotential height at 500hPa. It demonstrates the variability in the Z500 fields during GL blocking days, in particular along the edges of the block over Greenland pointing to variability in block size and exact location (and probably links to seasonal variations in jet position).

In comparison to other blocking identification methods, the weather regime perspective is based on a more conservative background by requiring a longer persistence (5 days). Using 10-day low-pass filtered fields of Z500 it removes some of the high-frequency synoptic activity, such that individual days within a GL life cycle would not necessarily represent a blocked pattern. However, considering blocking over Greenland as a weather regime allows to make statements of a blocked flow on the subseasonal time scale.

Nevertheless, there are large similarities between GL identified from a weather regime perspective versus classical blocking identification methods. A scatter plot showing the Greenland Blocking Index (GBI, Hanna et al., 2016) and the $I_{WR}$ of GL (Figure R1b) reveals a correlation coefficient of 0.545. Preece et al. (2022) identified 623 days of blocking in the summer for the period 1979-2019 based on the GBI. For the same period, 566 days were identified based on the weather regime definition. The differences might be the result of a slightly shorter persistence criterion used for the GBI blocking (4 days) compared to the one from the weather regime perspective (5 days). Given the large agreement between the GBI and other blocking identification methods such as by Davini et al. (2012) and Scherrer et al. (2006), we can assume that the results are robust.

We added the following sentence to the manuscript: "We find a high level of consistency of our regime-based GL definition with other blocking detection methods. For example, despite differences in persistence criteria, the $I_{WR}$ shows a positive correlation of 0.545 between to the Greenland Blocking Index (GBI) by (Hanna et al., 2016) and of the 623 GL days in summer found by Preece et al. (2022) during 1979-2019, our method identifies 566 days." (L. 173ff)

[Figure]

(a)            (b)

*Figure R1. (a) Standard deviation in geopotential height at 500hPa (shading) and Z500 anomalies (relative to 30-day running mean climatology, 1979-2019, in black contour lines) for all days assigned to an active GL life cycle in the period 1979-2021. Positive (negative) Z500 anomaly contours are shown in solid (dashed) lines with the intervals: 40, 60, 80, 100, 120, 140, 160gpm. (b) Scatter plot showing daily Greenland blocking index (GBI) from NOAA (source: https://psl.noaa.gov/ gcos_wgsp/Timeseries/GBI_UL/) vs. the $I_{WR}$ for GL. Blue (grey) points mark the GBI values during times of active (inactive) GL regime life cycles. The black star (square) shows the mean for the times of active (inactive) GL life cycles.*

L1: As currently written, the opening sentence reads like blocking over Greenland does not have impacts of surface weather over Greenland "Blocking over Greenland has substantial impacts on surface weather in particular over Europe and North America". I assume this is not what is meant so I would consider rephrasing this sentence.

Thank you for spotting this. We have changed the first sentence to "Blocking over Greenland stands out in comparison to blocking in other regions, as it favors accelerated Greenland Ice Sheet melting and has substantial impacts on surface weather in adjacent regions, particularly in Europe and North America." (Line 1f).

L65-80: Greenland blocks were also shown to be sensitive to upstream precursor cyclones and the upper-level wave pattern in Maddison et al. (2019). Consider including reference here.

Thank you, we found a spot in the introduction where an additional sentence on the study of Maddison et al. (2019) fits in well: "But also during autumn and spring, blocks over Greenland show sensitivities to upstream precursor cyclones and the upper-level wave pattern (Maddison et al., 2019)." (Line 68ff).

L126: could you add (half) a sentence here explaining why coarser resolution data is needed for the piecewise PV inversion?

Yes. We used coarser data than 0.5° because the PV Inversion was optimized for use with 1° data. We changed the sentence in the data section to "Spatially coarser data in the horizontal (1°) is selected as the piecewise PV inversion was optimized for a 1° resolution (see Teubler and Riemer, 2021)." (L. 126f)

L131: Given that ERA5 extends much further back now than 1979, was there a reason for only including these dates?

The reason why we decided not to extend our reanalysis period back to 1940 (ERA5) is the fact that prior to the satellite area (prior to 1979) the data quality in particular for free-tropospheric parameters declines (see e.g. Hersbach et al. EGU 2020 https://presentations.copernicus.org/EGU2020/EGU2020-10375_presentation.pdf, Slide 14). However, we rely on a high quality of the reanalysis for our process-oriented study and therefore decided to stick to the satellite era. In addition, we restrict the blocking definition to the same 40 years, as large-scale patterns have markedly decadal variability but were relatively constant in the period 1979-2019 (see the work of Dorrington et al., 2022). We decided not to add this justification in the manuscript.

L189: why are the threshold values in Fig. A1C positive?

We are sorry that we did not state this more precisely in the figure caption. The threshold is of course **negative**, but we decided for illustrative reasons to show the course of the absolute value of the threshold. However, to avoid confusion, we now show the evolution of the negative threshold in Figure A1C and updated the figure in the manuscript accordingly.

L226-L243: is this section missing a sentence on the physical interpretation of "Bnd"?

When we moved the part on the estimation of Bnd to the appendix, we should have left a sentence that explains the physical interpretation of Bnd and then directly points to the appendix. We have included the following sentences: "Equivalent to Hauser et al. (2023), we introduce the boundary term Bnd, which arises due to the fact that the PVAs$^-$ grow/shrink in size or deform leading to a change in the integration area A. A detailed documentation of the physical meaning of Bnd and of how the movement of the boundary of a PVAs$^-$ ($v_s$) is estimated is given in Appendix B." (L. 250ff).

L257 and elsewhere: many things are stated as being a "30-day running mean climatology". This could have several different meanings so a more precise definition should be included somewhere.

All 30-day running mean climatologies used in this work are produced as follows: (i) climatology for each time step, for example a mean of all fields on January 1 00 UTC 1979-2019; (ii) 30-day running mean climatology produced by taking the time steps +/- 15 days around a central date and perform a mean. We state this more explicitly in the manuscript when we first talk about centered running mean climatologies: "Note that all centered $n$-day running mean climatologies in this study are obtained by calculating the mean for each calendar time over the specified years and subsequently performing a mean centered on each calendar time taking into account the +/- $n/2$ surrounding days." (Line 135ff).

Fig.3: do these patterns look different if separated into seasons? A sentence could be added to highlight any differences or state they are similar.

We here provide a quick comparison of the patterns depending on the seasons in Figure R2 which shows the year-round, winter and summer GL pattern development during the days around GL onset. Independent of the season, a ridge and positive MSLP anomalies prevail over Europe and a trough linked to negative MSLP anomalies is located over the North Atlantic. Seasonal differences arise in the amplitude of PV anomalies despite using a 30-day centered running mean climatology with stronger anomalies in winter compared to summer, and the more wavy waveguide in winter. The pattern development in winter reminds of cyclonic RWB events from Europe towards Greenland before GL onset (e.g., Feldstein 2003, Michel and Rivière, 2011) and the respective dominance of the identified retrogression pathway in winter is clearly visible (Fig. R2h-j). We created a document with supplementary material for this manuscript and included Figure R2. In the manuscript, we added the following sentence: "A separate consideration of the development depending on the season indicates no fundamental, qualitative difference in the GL pattern development around onset, and that seasonal differences are dominated by the increased jet waviness and stronger anomalies in winter compared to summer (see Supplementary Figure S1). " (L. 300ff).

Fig.3: if the top row of Fig.3 is reproduced separately for the 'retrogression' and 'upstream' Greenland blocks, how different is it? Maybe the PV evolution in the days preceding the blocks are quite different for the different pathways leading to cancelation of features. Another sentence could be added on this if there are interesting differences.

We reproduced the top row of Fig. 3 separately for the two pathways and all three configurations (all, only retrogression, only upstream) as shown in Figure R3. The large-scale circulation pattern of GL develops differently for the two pathways, although in both cases a ridge is visible during the days before GL onset over Europe. However,

[Figure]

*Figure R2. Large-scale development of upper-tropospheric PV anomalies (shading, in PVU), Z500 (black contours), and anomalies of MSLP (red and blue contour lines). Anomalies are calculated relative to a 30-day running mean climatology spanning the period 1979-2019. Z500 contour intervals for first row: 5350, 5400, 5450, 5500, 5550 gpm, second row: 5250, 5300, 5350, 5400, 5450 gpm, and third row: 5450, 5500, 5550, 5600, 5650 gpm. Contour intervals for MSLP anomalies: +/- 2, 3, 4, 5 hPa for year-round, +/- 4, 5, 6, 7 for winter, and +/- 1, 2, 3, 4 for summer.*

while it is clear that the ridge over Europe propagates westward towards Greenland, a rapid development of a ridge takes place over the US East Coast for the upstream pathway cases. Hence, the patterns development in the days before GL onset shows distinct differences between the pathways. These differences are even larger than the differences between the seasons (winter vs. summer vs. year-round; cf. Figure R2), which are caused more by a difference in amplitude than by the pattern.

Teubler et al. (2023) showed snapshots of the PV dynamics for two modes of variability from their Eulerian perspective (their Fig. 8), but we decided not to make the focus of the pattern development on Eulerian maps in the manuscript here. We added Figure R3 to the supplementary material and included the following sentences to the end of Section 3.2: "The large-scale flow evolution around GL onset separated for the two pathways reflects the marked differences between the two pathways (Supplementary Figure S2): Except for the amplitude the evolution of the pattern differs more between the pathways (Supplementary Figure S2) than between seasons (Supplementary Figure S1)." (L. 344ff)

[Figure]

Figure R3. Large-scale development of upper-tropospheric PV anomalies (shading, in PVU), Z500 (black contours; contours levels: 5350, 5400, 5450, 5500, 5550 gpm), and anomalies of MSLP (red and blue contour lines; +/- 2, 3, 4, 5 hPa). Anomalies are calculated relative to a 30-day running mean climatology spanning the period 1979-2019.

L348: You state that LOW contributes to the amplification of onset PVA-s. Maybe I am confused, but an amplification of the PVA-s would be a negative tendency (as is seen in the total DIAG from day -2), whereas the tendency from LOW is positive throughout, I.e. acting against the block amplification? Also the text around L356. I assume I am interpreting this figure wrong as a positive PV tendency would not act to amplify the PVA-. Please clarify what you mean here.

In agreement with Teubler and Riemer (2016), Teubler and Riemer (2021), and Hauser et al. (2023), we decided to go with a sign convention that states **positive tendencies** as **amplifying contributions**. For negative PV anomalies, we multiplied the tendencies with -1, such that positive tendencies point to a strengthening of the PVA- amplitude and vice versa. We stated this more clearly in the manuscript when introducing the methodology (Section 2.3.2, L. 258ff), in the Caption of Figure 5 where we for the first time show integrated amplitude evolutions, and furthermore in the beginning of the main results sections (Section 3.3, L. 351ff; Section 4.2, L. 539f).

L440-453: does the increased WCB outflow at day-5 for the retrogression indicate that moist processes were also key in establishing the European/Scandinavian ridge that eventually becomes the Greenland block?

Yes, it does. We are currently working on a manuscript which looks into all four blocked regime types - Greenland blocking, Atlantic ridge, European blocking and Scandinavian

blocking - and investigates the differences in the dynamics and the role of regime transitions. The transition from Scandinavian blocking to Greenland blocking (or NAO-) is well known in the literature and we see that the retrogression of negative PV anomalies from Scandinavia to Greenland often co-occurs with blocking over Scandinavia before. However, for brevity here we focus more on the details of the Greenland Blocking life cycle alone. No chances have been made to the manuscript.

Section 4.3: you show the net effect of the different terms on the evolution of the maximum PVA-s. Have you looked at the net effect of the terms for all the PVA-s contributing to the Greenland blocks? How different would that look? (I am still confused why the PV tendencies are positive here when the onset of a block is associated with negative PV tendencies.)

Thanks for bringing this up. Actually, we haven't looked at this and one reason is that the number of PVAs- contributing to a Greenland blocking life cycle strongly varies. It would require a sophisticated approach to avoid overinterpretation just because of the number of PVAs⁻. As the timing of PVAs- located over Greenland varies within the life cycle, such an analysis would become very complex. However, we have done similar analyses for the onset PVAs⁻ and also for the decay stage (with decay PVAs⁻, respectively). Thereby, we have seen that the maximum PVAs⁻ are usually not the same anomalies as the onset and decay PVAs⁻. In the end, we decided not to do any additional analysis beyond this and no modifications to the manuscript have been made.

**Technical corrections:**

L95: adopting —> adapting? Corrected.

L225: missing ")" after "Equation 5" Corrected.

L291: is a word missing after "westward towards"? Do you mean "westward towards Greenland to build up a block"? Corrected. We decided to delete "towards".

L340: does DIAG not underestimate the amplitude change? OBS ends at a more negative value and therefore the change is greater than in DIAG?

DIAG overestimates the amplification of PVAs in the days before the onset shown in Figure 5 (note again the change in sign → positive means strengthening in amplitude, negative means weakening in amplitude of negative PV anomalies). However, we realized it is more an offset between DIAG and OBS than an overestimation of DIAG pointing to the lack of PV tendencies that would weaken the amplitude of PVAs-. This might be linked to the missing non-conservative tendency terms, which are shown to be dominated by weakening longwave radiative tendencies.

L466: the beginning decay phase —> the beginning of the decay phase? Changed.

L542: lost —> reduced Changed.

L550: remove "the" from "the sign" Changed.

L588: Sections 4.2 and 4.2 —> Sections 4.1 and 4.2?

Thanks for spotting this! We wanted to refer to the two subsections of Section 4.2, we therefore changed it to "... agrees well with the analyses in Section 4.2, indicating…." (L. 617f)

L629: on —> of Done.

**References provided by the reviewer:**

Barriopedro D, García-Herrera R, Trigo RM. Application of blocking diagnosis methods to general circulation models. Part I: a novel detection scheme. Clim Dyn. 2010;35(7–8):1373–91.

Maddison, J.W., Gray, S.L., Martínez-Alvarado, O. and Williams, K.D. (2019) Upstream cyclone influence on the predictability of block onsets over the Euro-Atlantic region. Monthly Weather Review, 147, 1277–1296.

―――――――――――――――――――――――――――――――――――――――――――

**Reviewer 2**

―――――――――――――――――――――――――――――――――――――――――――

This study identifies consistent life cycles of Greenland blocking with objective onset, maximum and decay stages in ERA5 reanalysis data from 1979–2021. The periods of blocking are identified from the perspective of 7 weather regimes in the North Atlantic-European region, and a weather regime index computed to quantify the similarity of an instantaneous geopotential height field to one of these regimes. These indices are used to define onset and decay times and thus regime life cycles, of at least 5 days. One of these weather regimes is an anticyclonic type, Greenland Blocking (GL). This method identifies 177 GL episodes in the data set.

The study then employs a quasi-Lagrangian potential vorticity (PV) perspective to track negative, upper-tropospheric PV anomalies (PVAs-) associated with the block, to quantify the contribution to the evolution of the amplitude of these anomalies from different physical processes, including dry and moist dynamics.

The authors find that the PVAs- linked to GL do not form locally, but follow two distinct pathways: upstream and retrogression; these pathways are identified based on two

areas east and west of the centre of mass longitude of the year-round composite vertically averaged negative PV anomaly. Upstream PVAs- propagate north-eastward towards Greenland from the northern United States, whereas retrogression PVAs- are characterised by a north-westward propagation against the mean flow from norther Europe. There are more differences in the development of PVAs- between these two pathways than there is between seasons.

An Eulerian method based on convolutional neural networks is used to identify warm conveyor belts (WCBs) and identify inflow in the lower troposphere, ascent in the mid-troposphere, and outflow in the upper troposphere.

Using divergent PV tendencies, the study finds that moist processes play a key role in the amplification of PVAs- regardless of pathway. The authors further link the increase in PVAs- amplitude to WCB activity over the North Atlantic. Upper-level wave dynamics and baroclinic interaction also play a role.

The article is very well written with high quality figures, and fits well within the scope of WCD. The analysis techniques are appropriate, and make use of recent novel methods to further the understanding of the formation of GL. The conclusions reached are supported by the analysis, and there is the potential for further valuable study based on these results, e.g. the representation of the identified mechanisms in models. The results will be of broad interest, and I recommend the publication of this manuscript in the journal subject to some minor comments.

We thank the reviewer for this positive feedback and the to-the-point comments.

**Minor comments:**

1. How do the frequency and seasonality of Greenland blocking events found using the weather regime perspective, compare over a similar time period with events found using other blocking indices? For example, in Parker et al (2018), 26 events were found in a 10-year period from 2006-2015.; this study finds 177 events in a 43-year period. A short discussion of this would frame the results in the context of other studies.

This is a valid comment and we are happy to provide more comparisons on how our Greenland blocking episodes compare to Greenland blocking defined by more common blocking indices. With our regime-based approach, we only find 7 Greenland blocking events compared to the 26 events found by Parker et al. (2018) in the period from 2006 - 2015 (DJF). These differences can be caused by the following factors: (1) instantaneous fields in Parker et al. (2018)  vs. 10-day low pass filtering in our method, (2) varying difference of the Greenland blocking region, (3) threshold in minimum duration of blocking, (4) year-round vs. seasonal classification. Figure R2 provides a comparison of summer Greenland blocking identification by comparing different blocking identification methods: Pelly and Hoskins index, Greenland blocking index (GBI, see above answer to R1 comment 1), adjusted version of the GBI (GBI-Z), and our regime-based definition of Greenland blocking episodes. Although some periods are not picked up from the weather regime perspective there is a large overlap. Please see also the reply to

[Figure]

*Figure R4: Comparison of different blocking indices identifying blocking over Greenland (Figure adapted from Preece et al., 2022, their Figure S2). Blue bar on the top of the Figure shows the identified Greenland blocking regime life cycles in this study.*

Reviewer 1, where we compared the GBI index by Hanna et al. (2016) to our Greenland blocking episodes from a weather regime perspective (Fig. R1b).

In summary, less Greenland blocking episodes are found from our perspective compared to other blocking indices. The most obvious reason stems from the fact that the regime perspective requires some persistence. However, if a persistent flow situation is identified, it allows for some transience if the pattern becomes intermittently weaker and reestablishes (e.g. in June/July 2012), such that single short blocking episodes identified by other algorithms are sometimes seen as one long period (see 2012 in Figure R4). We have added a sentence in the manuscript (see also first reply to Reviewer 1): "We find a high level of consistency of our regime-based GL definition with other blocking detection methods. For example, despite differences in persistence criteria, the $I_{WR}$ shows a positive correlation of 0.545 between to the Greenland Blocking Index (GBI) by (Hanna et al., 2016) and of the 623 GL days in summer found by Preece et al. (2022) during 1979-2019, our method identifies 566 days." (L. 173ff)

2. The GL life cycle varies in length from 5 days to more than a month. Are there any major differences in the evolution of the PVAs- between shorter- and longer-lived GLs?

We have looked at differences in the evolution of PVAs- between short-lived and long-lived GL episodes with a focus on the processes linked to the maximum PVAs- *within* the regime life cycles (from onset to decay). We found larger divergent and baroclinic PV tendency contributions to the overall amplification of PVAs- from onset to

[Figure]

(a)                                                                    (b)

Figure R5: (a) Scatterplot of diagnosed (x-axis) and observed (y-axis) amplitude change for all considered time steps (-6 days → +2 days) around GL onset. Coloring shows which data points are not filtered out when applying a certain threshold (thresholds in legend are in $10^7$ PVU m$^2$ s$^{-1}$, e.g. 5 → 5 · $10^7$ PVU m$^2$ s$^{-1}$). The percentage in the legend shows how many data points are kept. (b) Amplitude evolution (diagnosed in solid, observed in dashed) for the different thresholds applied. Note again that positive values point to a strengthening of the PVA$^-$ amplitude (we multiplied it by -1!) and that the time steps +/- 12 hours are taken into account for slight smoothing.

maximum for longer GL life cycles, however the differences are rather small. We have therefore decided against an analysis of short and long life cycles of GL in the manuscript and did not make any changes in the manuscript text.

3. Lines 334-342: questionable time steps when OBS and DIAG exhibit very different values have been filtered out. Can you motivate the choice of the condition for this filtering, and quantify the "large fraction" of values that are still included in the composite?

At the time of the submission, the threshold was chosen subjectively. The main goal was to filter out the data for times when the diagnosed (DIAG) and observed (OBS) amplitude changes are very different, which means that DIAG fails to describe the actual amplitude evolution. Reasons for the disagreement between OBS and DIAG are discussed in the appendix. Importantly, because of the lack of the non-conservative PV tendencies in this work, DIAG always overestimates the amplification of PVAs- and underestimate the amplitude weakening of PVAs$^-$ (cf.; Fig. 10b,c in Hauser et al., 2023; Fig. 7a,c in Teubler and Riemer, 2021). We therefore allow a certain spread between DIAG and OBS. When choosing the threshold of 2.5 $10^7$ PVU m$^2$ s$^{-1}$, in total 66% of all values are kept. In the mean, 111 mean-isentropic values go into the year-round

[Figure]

*Figure R6: Amplitude evolution of PVAs- split up into different seasons (extended winter/ summer) or into the two pathways (retrogression/upstream). Solid (dashed) lines represent the diagnosed (observed) amplitude change. Note that the 2.5 · 10⁷ PVU m² s⁻¹ criterion has been applied.* Note again that positive values point to a strengthening of the PVA⁻ amplitude (we multiplied it by -1!) and that the time steps +/- 12 hours are taken into account for slight smoothing.

composite of possible 177 values. Furthermore, it must be noted that (1) not all PVAs⁻ exist at every time in the composite (-6 days to +2 days), and (2) splitting and merging does occur and leads to big jumps in OBS, which DIAG can follow only to a certain degree.

Figure R5a shows a scatter plot of OBS-DIAG pairs and the coloring displays which values are kept. In addition, we show in Figure R5b how the mean amplitude evolution changes depending on the threshold that is used to filter out questionable time steps. If no threshold value is selected, the mean amplitude change assumes unphysical values (black solid line in Figure R5b), which are most likely caused by individual abrupt amplitude changes due to splitting/merging. If we select a higher threshold value, the sample size will be smaller (Figure R5a) but the agreement between DIAG and OBS will be better, the smaller the threshold (Figure R5b). In order not to lose an excessive number of data points, we ultimately decided subjectively in favor of the threshold value 3.5 · 10⁷ PVU m² s⁻¹, whereby it is clear from Figure R5b that the threshold value is only weakly sensitive and the amplitude change remains qualitatively the same.

We have extended the previous short paragraph to: "The subjectively determined condition |OBS -- DIAG| < 2.5 10⁷ PVU m² s⁻¹ ensures that time steps when DIAG deviates very strongly from OBS are excluded, but at the same time a large fraction of values are still included in the composite (~66%). The threshold value is only weakly sensitive and the amplitude change remains qualitatively the same (not shown). With the applied filtering, DIAG still slightly overestimates the amplification and underestimates the amplitude weakening of the onset PVAs⁻ (Figure 5a), but the temporal variations of the curves are very similar and the agreement is thus sufficiently good for our analysis." (L. 360ff).

[Figure]

*Figure R7: Composites centered on the position (center of mass) of onset PVAs⁻ for selected times relative to GL onset (columns) and for the two pathways (year-round) and seasons (independent of pathways) separately (rows) showing the occurrence of different WCB stages (shading) and VAPV' (black contours). Colored shading indicates the frequency of WCB inflow in the lower troposphere (blue, from 0.02 to 0.04 in steps of 0.005), WCB ascent in the mid-troposphere (green, from 0.02 to 0.045 in steps of 0.005), and WCB outflow in the upper troposphere (red, 0.06 to 0.16 in steps of 0.02). Solid and dashed black contours illustrate the positive and negative VAPV' (500-150hPa), respectively. The contour levels displayed are [-1.3, -1.0, -0.7, -0.4, -0.1, 0.1, 0.4, and 0.7] PVU. PV tendencies of DIV$_{div}$ are shown in gold with contour levels of [-6, -8, -10, -12, -14] 10⁶ PVU m² s⁻¹. All fields shown are smoothed by a Gaussian filter with sigma = 2.*

4. You state that the differences in the development of PVAs⁻ is greater between the two pathways than between seasons. Do you have any figures readily available to show this? – this is purely out of curiosity, but perhaps some supplementary material would be

of interest to readers. The seasonality in the contribution of WCBs/moist processes would be interesting to see.

Many thanks for raising this. We share here two additional analyses (Figures R6 and R7) to discuss this in more detail and to quantify more precisely what we mean by this statement. Figure R6 shows the amplitude evolution around onset for the two pathways (year-round) and for the two seasons independent of pathways. All scenarios share the amplitude strengthening before the onset and amplitude weakening after onset. The amplification of onset PVAs$^-$ is stronger in winter compared to summer, in particular 2 days before the GL onset (red and dark red lines). Figure R7 shows centered composites on the onset PVAs$^-$ center of mass and clearly indicates that the strength/frequency in WCB activity is strongly dominated by seasonality (more WCB activity in winter compared to summer, cf. Madonna et al., 2014). As a result, the strong peak in amplification in winter two days before the onset (Figure R6, dark red line) is linked to increased WCB activity and stronger negative tendencies of DIV$_{div}$ (Figure R7, third row).

Aside from the amplitude of the amplification, the development of the amplification in time shows distinct differences between the pathways. We identified amplification peaks with a clear time lag between the two pathways and a generally opposing behavior (in particular in the period -6 to -2 days). This suggests a clearer difference in the underlying dynamics between the two pathways than between the two seasons. Please note that the seasonality also plays an indirect role due to the dominance of the retrogression pathway in winter.

We included both figures (Figure R6 and Figure R7) in the supplementary material of the manuscript and added the following paragraph to the manuscript: "The seasonal stratification of WCB activity around GL onset (Supplementary Figure S4) highlights the known marked differences in WCB detection in winter compared to summer (cf. Madonna et al., 2014). Still the qualitative picture is similar independent of the season with enhanced WCB activity on the upstream flank of the incipient block prior to and maximized just before onset." (L. 487ff). Furthermore, we adjusted the summary at the end of the amplitude evolution discussion to "We found that seasonal differences are mainly caused by lower baroclinicity in summer compared to winter resulting in a distinct higher amplification of onset PVAs$^-$ in winter before the onset. However, the partitioning into the two pathways reveals fundamental differences in the large-scale flow patterns in which the onset PVAs$^-$ are embedded, as well as in the amplitude evolution, where we found distinct time lags between the maximum amplification and even an opposite qualitative evolution." (L. 437ff), and in the summary of the manuscript we clarified it again in more detail by adding half a sentence to: "Although previous studies recommend splitting up process-based analyses into seasons, we found more distinct differences in the dynamics between the pathways than between individual seasons, leaving aside pure differences in amplitude strength that arise due to weaker baroclinicity in summer." (L. 645ff)

**Edits:**

1. Line 95 – "adopting" should be "adapting". Done.
2. Line 225 – missing ) after Equation 5. Done.
3. 3 caption – for the bottom row, what is the contour interval for the orange and green contours? – is it the same as the grey shading? Thank you, we have added the missing information on the contours.
4. Line 291 – "propagated westward towards" – towards where? Removed "towards" (see Reviewer 1 comment, too).
5. Lines 318-319 – "the chosen period of three days before onset is sufficient, as this is the time period with the largest differences in the propagation of onset PVAs- ": perhaps note here that this is shown by the white crosses in Fig 4 which mark the point of three days before onset (-96 hrs). This motivates the choice of the centre of mass location to determine the pathway classification, and makes it clearer for the reader. Done.
6. Line 337 – Section B should be Appendix B. Thank you! We corrected it.
7. Line 340 – "DIAG somewhat over-estimate" – should this be "DIAG somewhat under-estimates"? See answer to comment of reviewer 1 above.
8. Line 466 – "the beginning decay phase" – perhaps rephrase as "the beginning of the decay phase". Done.
9. Line 521 - "indicating a starting decrease in max PVA- amplitude" - perhaps rephrase to "indicating the beginning of a decrease in maximum PVA- amplitude". Done.
10. Line 542 – "the ridge further weakened and lost in northward extent" – perhaps rephrase to "the ridge further weakened and contracted to the south". Changed, see Reviewer comment 1.
11. Line 543 – "an even retrograde behaviour of the ridge" – perhaps rephrase to "a possible retrogression of the ridge" or the like. Thank you, we changed it accordingly.
12. Line 550-551 – "which switches the sign" should be "which switches sign". Done.
13. Line 581 – after the first sentence of this paragraph, reference Figure 12, right side. Done.
14. Line 588 – "Sections 4.2 and 4.2"?? Changed! It should just be Section 4.2 (see also Reviewer 1)
15. Line 629 – "are on minor" should be "are of minor". Corrected.
16. Line 654 – "Taken the two perspectives…" should be "Taking the two perspectives.." Done.

**References added by the authors**

Davini, P., Cagnazzo, C., Gualdi, S., & Navarra, A. (2012). Bidimensional diagnostics, variability, and trends of northern hemisphere blocking. Journal of Climate, 25(19), 6496–6509. https://doi.org/10.1175/JCLI-D-12-00032.1

Dorrington, J., & Strommen, K. J. (2020). Jet Speed Variability Obscures Euro-Atlantic Regime Structure. Geophysical Research Letters, 47(15). https://doi.org/10.1029/2020GL087907

Feldstein, S. B. (2003). The dynamics of NAO teleconnection pattern growth and decay. Quarterly Journal of the Royal Meteorological Society, 129(589 PART C), 901–924. https://doi.org/10.1256/qj.02.76

Michel, C., & Rivière, G. (2011). The link between rossby wave breakings and weather regime transitions. Journal of the Atmospheric Sciences, 68(8), 1730–1748. https://doi.org/10.1175/2011JAS3635.1

Hanna, E., Cropper, T. E., Hall, R. J., & Cappelen, J. (2016). Greenland Blocking Index 1851–2015: a regional climate change signal. International Journal of Climatology, 36(15), 4847–4861. https://doi.org/10.1002/joc.4673

Hauser, S., Teubler, F., Riemer, M., Knippertz, P., & Grams, C. M. (2023). Towards a holistic understanding of blocked regime dynamics through a combination of complementary diagnostic perspectives. Weather and Climate Dynamics, 4(2), 399–425. https://doi.org/10.5194/wcd-4-399-2023

Maddison, J. W., Gray, S. L., Martínez-Alvarado, O., & Williams, K. D. (2019). Upstream cyclone influence on the predictability of block onsets over the Euro-Atlantic region. Monthly Weather Review, 147(4), 1277–1296. https://doi.org/10.1175/MWR-D-18-0226.1

Madonna, E., Wernli, H., Joos, H., & Martius, O. (2014). Warm conveyor belts in the ERA-Interim Dataset (1979-2010). Part I: Climatology and potential vorticity evolution. Journal of Climate, 27(1), 3–26. https://doi.org/10.1175/JCLI-D-12-00720.1

Preece, J. R., Wachowicz, L. J., Mote, T. L., Tedesco, M., & Fettweis, X. (2022). Summer Greenland Blocking Diversity and Its Impact on the Surface Mass Balance of the Greenland Ice Sheet. Journal of Geophysical Research: Atmospheres, 127(4). https://doi.org/10.1029/2021JD035489

Parker, T., Woollings, T., & Weisheimer, A. (2018). Ensemble sensitivity analysis of Greenland blocking in medium-range forecasts. Quarterly Journal of the Royal Meteorological Society, 144(716), 2358–2379. https://doi.org/10.1002/qj.3391

Scherrer, S. C., Croci-Maspoli, M., Schwierz, C., & Appenzeller, C. (2006). Two-dimensional indices of atmospheric blocking and their statistical relationship with winter climate patterns in the Euro-Atlantic region. International Journal of Climatology, 26(2), 233–249. https://doi.org/10.1002/joc.1250

Teubler, F., & Riemer, M. (2016). Dynamics of Rossby wave packets in a quantitative potential vorticity-potential temperature framework. Journal of the Atmospheric Sciences, 73(3), 1063–1081. https://doi.org/10.1175/JAS-D-15-0162.1

Teubler, F., & Riemer, M. (2021). Potential-vorticity dynamics of troughs and ridges within Rossby wave packets during a 40-year reanalysis period. Weather and Climate Dynamics, 2(3), 535–559. https://doi.org/10.5194/wcd-2-535-2021

---

## Referee Report (RR1)

I wish to thank the authors for their detailed responses to the first round of reviewer comments. All of my concerns have been addressed and I therefore recommend the paper be published in its current form.